# Combining genotypes and T cell receptor distributions to infer genetic loci determining V(D)J recombination probabilities

Magdalena L Russell[1,2]*, Aisha Souquette[3,4], David M Levine[5], Stefan A Schattgen[3], E Kaitlynn Allen[3], Guillermina Kuan[6,7], Noah Simon[5], Angel Balmaseda[6,7], Aubree Gordon[8], Paul G Thomas[3], Frederick A Matsen IV[1,9,10,11]*†, Philip Bradley[1,12]*†

[1]Computational Biology Program, Fred Hutch Cancer Research Center, Seattle, United States; [2]Molecular and Cellular Biology Program, University of Washington, Seattle, United States; [3]Department of Immunology, St. Jude Children's Research Hospital, Memphis, United States; [4]Department of Microbiology, Immunology, and Biochemistry, University of Tennessee Health Science Center, Memphis, United States; [5]Department of Biostatistics, University of Washington, Seattle, United States; [6]Centro Nacional de Diagnóstico y Referencia, Ministry of Health, Managua, Nicaragua; [7]Sustainable Sciences Institute, Managua, Nicaragua; [8]Department of Epidemiology, University of Michigan, Ann Arbor, United States; [9]Department of Genome Sciences, University of Washington, Seattle, United States; [10]Department of Statistics, University of Washington, Seattle, United States; [11]Howard Hughes Medical Institute, Seattle, United States; [12]Institute for Protein Design, Department of Biochemistry, University of Washington, Seattle, United States

*For correspondence:
magruss@uw.edu (MLR);
matsen@fredhutch.org (FAM);
pbradley@fredhutch.org (PB)

†These authors contributed equally to this work

**Abstract** Every T cell receptor (TCR) repertoire is shaped by a complex probabilistic tangle of genetically determined biases and immune exposures. T cells combine a random V(D)J recombination process with a selection process to generate highly diverse and functional TCRs. The extent to which an individual's genetic background is associated with their resulting TCR repertoire diversity has yet to be fully explored. Using a previously published repertoire sequencing dataset paired with high-resolution genome-wide genotyping from a large human cohort, we infer specific genetic loci associated with V(D)J recombination probabilities using genome-wide association inference. We show that V(D)J gene usage profiles are associated with variation in the *TCRB* locus and, specifically for the functional TCR repertoire, variation in the major histocompatibility complex locus. Further, we identify specific variations in the genes encoding the Artemis protein and the TdT protein to be associated with biasing junctional nucleotide deletion and N-insertion, respectively. These results refine our understanding of genetically-determined TCR repertoire biases by confirming and extending previous studies on the genetic determinants of V(D)J gene usage and providing the first examples of *trans* genetic variants which are associated with modifying junctional diversity. Together, these insights lay the groundwork for further explorations into how immune responses vary between individuals.

## Editor's evaluation

This study demonstrates that genetic differences in areas of the genome outside the regions that encode the TCR genes can affect the molecular properties of the TCRs that are made by somatic

recombination. This paper will be of interest to a broad swathe of immunologists who study such variable lymphocyte receptors. It combines several large datasets in an extremely statistically rigorous analysis, producing results consistent with but substantially expanding upon the prior knowledge of the field.

## Introduction

Receptor proteins on the surfaces of T cells are an essential component of the cell-mediated adaptive immune response in humans. Cells throughout the body regularly present protein fragments, known as antigens, on cell-surface molecules called major histocompatibility complex (MHC). Each T cell expresses a randomly-generated T cell receptor (TCR) which can bind the MHC-bound peptide and, if necessary, initiate an immune response. As part of this immune response, a T cell will proliferate and subsequent clones of that T cell will inherit the same antigen-specific TCR. Over time, the collection of all TCRs in an individual (the TCR repertoire) will dynamically summarize their previous immune exposures (*Woodsworth et al., 2013*).

To appropriately defend against a wide array of foreign pathogens, each individual has a highly diverse TCR repertoire. To generate diverse and functional TCRs, T cells combine a random generation process called V(D)J recombination with a selection process for proper expression and MHC recognition. Each TCR is composed of an $\alpha$ and a $\beta$ protein chain which are both generated through V(D)J recombination. In the $\beta$ chain, the recombination process proceeds by randomly choosing from a pool of V-gene, D-gene, and J-gene segments of the germline T cell receptor beta (*TCRB*) locus over a series of steps. First, the intervening chromosomal DNA between a randomly chosen D- and J-gene is removed to form a hairpin loop at the end of each gene (*Gellert, 1994*; *Fugmann et al., 2000*; *Schatz and Swanson, 2011*). Next, these hairpin loops are nicked open, often asymmetrically, by the Artemis-DNA-PKcs protein complex to create overhangs at the ends of each gene (*Weigert et al., 1978*; *Moshous et al., 2001*; *Ma et al., 2002*; *Lu et al., 2007*; *Zhao et al., 2020*). Depending on the location of the nick, the single-stranded overhang can contain short inverted repeats of gene terminal sequence known as P-nucleotides (*Nadel and Feeney, 1995*; *Gauss and Lieber, 1996*; *Nadel and Feeney, 1997*; *Jackson et al., 2004*). From here, nucleotides may be deleted from the gene ends through an incompletely understood mechanism suggested to involve Artemis (*Feeney et al., 1994*; *Nadel and Feeney, 1995*; *Nadel and Feeney, 1997*; *Jackson et al., 2004*; *Gu et al., 2010*; *Zhao et al., 2020*). This nucleotide trimming can remove traces of P-nucleotides (*Gauss and Lieber, 1996*; *Srivastava and Robins, 2012*). Next, non-templated nucleotides, known as N-insertions, can be added between the gene segments by the enzyme terminal deoxynucleotidyl transferase (TdT) (*Kallenbach et al., 1992*; *Gilfillan et al., 1993*; *Komori et al., 1993*). Once the nucleotide addition and deletion steps are completed, the gene segments are ligated together. The process is then repeated between this D-J junction and a random V-gene segment to generate a complete TCRβ protein chain. After the β chain has been generated, a similar α chain recombination proceeds, although without a D-gene, to complete the TCR. Following the generation process, each completed TCR undergoes a selection process in the thymus to limit autoreactivity and ensure its ability to correctly bind peptide antigens presented on a specific MHC molecule (*Goldrath and Bevan, 1999*; *Thomas and Crawford, 2019*).

TCR repertoires vary between individuals and are a complicated tangle of genetically determined biases and immune exposures. Disentangling these factors is essential for understanding how our diverse repertoires support a powerful immune response. Previous efforts to unravel the genetic and environmental determinants governing TCR repertoire diversity have been highly impactful despite lacking high-throughput TCR repertoire sequencing data (*Sharon et al., 2016*; *Gao et al., 2019*) and/or high-resolution genotype data (*Rubelt et al., 2016*; *Emerson et al., 2017*; *Gao et al., 2019*; *Krishna et al., 2020*). For example, it has been shown that variation in the MHC locus biases TCR V(D)J gene usage (*Sharon et al., 2016*; *Gao et al., 2019*) and has been associated with clusters of shared receptors in response to Epstein-Barr virus epitope (*DeWitt et al., 2018*). Other studies have reported biases in V(D)J gene usage (*Zvyagin et al., 2014*; *Qi et al., 2016*; *Rubelt et al., 2016*; *Pogorelyy et al., 2018*; *Tanno et al., 2020*; *Fischer et al., 2021*), N-insertion lengths (*Rubelt et al., 2016*), and repertoire similarity in response to acute infection (*Qi et al., 2016*; *Pogorelyy et al., 2018*) for monozygotic twins. While this work clearly illustrates that genetic similarity implies TCR repertoire

**Table 1.** Discovery cohort demographics.

|  |  | Count |
|---|---|---|
| Sex | Female | 179 |
|  | Male | 197 |
|  | Unknown | 22 |
| Age (in years) | < 10 | 12 |
|  | 11–20 | 11 |
|  | 21–30 | 48 |
|  | 31–40 | 70 |
|  | 41–50 | 103 |
|  | 51–60 | 70 |
|  | > 60 | 22 |
|  | Unknown | 62 |
| Ancestry-informative PCA cluster (see Materials and methods) | "African"-associated | 8 |
|  | "Asian"-associated | 23 |
|  | "Caucasian"-associated | 322 |
|  | "Hispanic"-associated | 30 |
|  | "Middle Eastern"-associated | 5 |
|  | "Native American"-associated | 10 |
| CMV serostatus | Positive | 171 |
|  | Negative | 204 |
|  | Unknown | 23 |
| Total |  | 398 |

The online version of this article includes the following source data for table 1:

**Source data 1.** Subjects map from the original self-identified ancestry groups to ancestry-informative PCA clusters (see Materials and methods).

similarity, the extent to which specific variations are associated with V(D)J recombination probabilities has not been fully explored.

In this paper, we directly address the question of how an individual's genetic background influences their V(D)J recombination probabilities using large human discovery and validation cohorts for which both TCR immunosequencing data (*Emerson et al., 2017*; *DeWitt et al., 2018*) and genotyping data (*Martin et al., 2020*) are available. With the goal of identifying statistically significant associations between single nucleotide polymorphisms (SNPs) and TCR repertoire features of interest using these novel, paired datasets, we treat analysis as a genome-wide association (GWAS) inference with many outcomes. Our results suggest that MHC and *TCRB* loci variations have an important role in determining the V(D)J gene usage profiles of each individual's repertoire. At the junctions, we demonstrate that variations in the genes encoding the Artemis protein and the TdT protein are associated with biasing V- and J-gene nucleotide deletion and V-D and D-J-junction N-insertion, respectively.

## Results

### Discovery cohort data description

We worked with paired SNP array and TCRβ-immunosequencing data representing 398 individuals and over 35 million SNPs and/or indels (*Table 1*). TCR sequences can be separated into those that code for a complete, full-length peptide sequence (which we will call 'productive' rearrangements)

**Table 2.** We inferred the associations between genome-wide variation and many different TCR repertoire features for productive and non-productive TCR sequences, separately.

For each TCR repertoire feature, we considered the significance of associations using a Bonferroni-corrected threshold established to correct for each TCR feature subtype, the two TCR productivity types, and the total number of SNPs tested (described in detail in Methods).

| Repertoire feature (significance threshold) | Model type | Feature subtype | Productivity | Significant association |
|---|---|---|---|---|
| V(D)J gene usage $(5.09 \times 10^{-11})$ | simple | Each of 60 V-genes | Productive | Yes, for 36 V-genes |
| | | | Non-productive | Yes, for 26 V-genes |
| | | Each of 2 D-genes | Productive | Yes, for both D-genes |
| | | | Non-productive | Yes, for both D-genes |
| | | Each of 14 J-genes | Productive | Yes, for 7 J-genes |
| | | | Non-productive | Yes, for 8 J-genes |
| Amount of nucleotide trimming $(9.68 \times 10^{-10})$ | gene-conditioned | V-gene trimming | Productive | Yes |
| | | | Non-productive | Yes |
| | | 5' end D-gene trimming | Productive | No |
| | | | Non-productive | No |
| | | 3' end D-gene trimming | Productive | No |
| | | | Non-productive | No |
| | | J-gene trimming | Productive | Yes |
| | | | Non-productive | Yes |
| Number of N-insertions $(1.94 \times 10^{-9})$ | simple | V-D-gene N-insertions | Productive | No |
| | | | Non-productive | Yes |
| | | D-J-gene N-insertions | Productive | No |
| | | | Non-productive | Yes |

and 'non-productive' rearrangements that do not. Non-productive sequences can arise during TCR generation steps if the V- and J-genes are shifted into different reading frames or a premature stop codon is introduced in the junction region. A non-productive rearrangement can be sequenced as part of the repertoire when a recombination event on one of a T cell's two chromosomes fails to create a functional receptor, followed by a successful recombination event on the other chromosome. Because these non-productive sequences do not generate proteins that participate in the T cell selection process within the thymus, they should not be subject to functional selection (*Robins et al., 2010*; *Murugan et al., 2012*). As such, their recombination statistics should reflect only the V(D)J recombination generation process which occurs before the stage of thymic selection.

In the data cohort of 398 individuals, an average of 235,054 unique TCRβ-chain nucleotide sequences were sequenced per individual. Within each individual repertoire, roughly 18% of sequences were classified as 'non-productive'. Thus, we can analyze the productive and non-productive sequences separately to distinguish between TCR generation and selection effects within each TCR repertoire. Specifically, we inferred the associations between genome-wide variation and V(D)J gene usage of each V-, D-, and J-gene, the extent of TCR nucleotide trimming, the number of TCR N-insertions, and the fraction of non-gene-trimmed TCRs containing P-nucleotides for both productive and non-productive sequences (*Table 2*).

## *TCRB* and MHC locus variation is associated with V-, D-, and J-gene usage frequency

To quantify the effect of SNPs on the expression of various V-, D-, and J-genes during V(D)J recombination, we designed a fixed effects model to assess the relationship between SNP genotype and gene frequency across all individuals. We fit this 'simple model' for each different V-, D-, and J-gene in our paired dataset.

Because of the potential for population-substructure-related effects to inflate associations between each SNP and gene usage frequency, we incorporated ancestry-informative principal components (*Conomos et al., 2015*) based on the SNP genotypes for a subset of representative subjects as covariates in each model (see Materials and methods for details). Diagnostic statistics show that this bias correction is sufficient (*Figure 5—source data 3*).

With these methods, we consider the significance of associations at a Bonferroni-corrected whole-genome p-value significance threshold of $5.09 \times 10^{-11}$ (see Materials and methods). Using this conservative threshold, we identified 9152 significant associations between the frequency of various V-, D-, and J-genes and the genotype of SNPs genome-wide (*Figure 1* and *Figure 1—source data 1*). Of these significant associations, 7096 were located within the *TCRB* locus for both productive and non-productive TCRs. The *TCRB* gene locus encodes the variable V-, D-, and J-gene segments which are recombined during V(D)J recombination. In our dataset, there are 60 V-genes, 2 D-genes, and 14 J-genes uniquely expressed. As we would expect, we find that the expression of many of these genes is associated with variation in the *TCRB* locus (*Figure 2*). For the significantly associated *TCRB* locus SNPs, the median association effect magnitude was largest for the expression of TRBD1 (median effect size = –0.038) followed by the expression of TRBD2 (median effect size = 0.035) and the expression of TRBV28 (median effect size = 0.019) all in productive TCRs (*Figure 1—figure supplement 1*). Variation in the *TCRB* locus is most significantly associated (smallest p-value) with expression of the gene *TRBV28* within both productive ($P = 1.41 \times 10^{-164}$) and non-productive ($P = 1.94 \times 10^{-146}$) TCRβ chains. We identified the largest number of significant associations between variation in the *TCRB* locus and expression of the gene *TRBV7-3* within productive TCRβ chains (232 significant associations) and the gene *TRBJ1-2* within non-productive TCRβ chains (290 significant associations).

Beyond the *TCRB* locus, we also identified 1242 significant SNP associations within the major histocompatibility complex (MHC) locus. MHC proteins act by presenting self and foreign peptides to TCRs for inspection. Because of this important role in the functionality of T cells, the TCR-MHC interaction is important for thymic selection. We observe the expression of 12.1% of V-genes for productive TCRs to be associated with variation in the MHC locus. For the significantly associated MHC locus SNPs, the median association effect magnitude was largest for the expression of TRBV4-1 (median effect size = –0.004) followed by the expression of TRBV10-3 (median effect size = 0.0033) (*Figure 1—figure supplement 2*). This associated MHC locus variation is located within sequences which code for canonical, peptide-presenting MHC proteins. For example, the eight most significantly associated SNPs were located within the *HLA-DRB1* gene within the MHC locus. These top SNPs were all associated with the expression of the gene *TRBV10-3* within productive TCRs. As expected, the expression of V-genes for non-productive TCRs is not associated with variation in the MHC locus. Likewise, the expression of D- and J-genes for both productive and non-productive TCRs is not associated with variation in the MHC locus. These results refine and extend associations found in previous work (*Sharon et al., 2016*; *Gao et al., 2019*).

We observed just one other long-range association region, in addition to the MHC locus, located in proximity to the *ZNF443* and *ZNF709* loci on chromosome 19. Both of these zinc finger proteins contain KRAB-domains and, thus, likely act as transcriptional repressors (*Witzgall et al., 1994*). In this region, we observe 138 significant SNP associations for the expression of the V-gene *TRBV24-1*. Of these 138 SNP associations, 76 were associations for *TRBV24-1* expression in non-productive TCRs and 62 were associations for *TRBV24-1* expression in productive TCRs. Significant association between variation near the *ZNF443* locus and expression of *TRBV24-1* in productive TCRs was also noted previously (*Sharon et al., 2016*). Because the associations observed here are strongest for non-productive TCRs, this chromosome 19 variation likely influences gene usage during TCR generation steps, as opposed to selection. Variation in proximity to the *ZNF443* and *ZNF709* loci may alter the resulting zinc finger proteins and lead to differential transcriptional repression of a site near *TRBV24*. Because the transcription of unrearranged gene segments influences their recombination potential

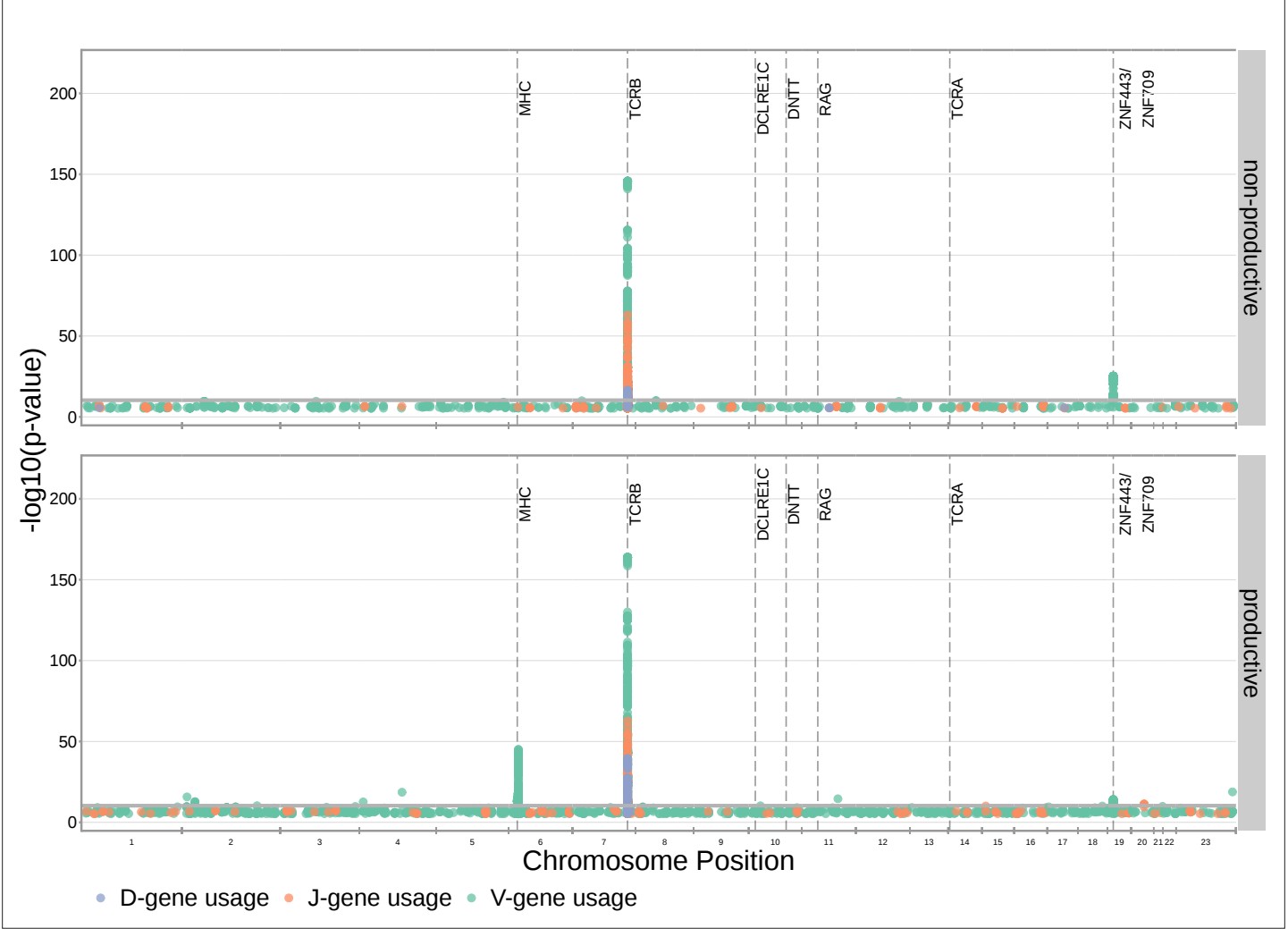

**Figure 1.** Many strong associations are present between V-, D-, and J-gene usage frequency and various SNPs genome-wide for both productive and non-productive TCRs. The most significant SNP associations for the frequency of each of the 60 V-genes, 2 D-genes, and 14 J-genes are located within the *TCRB* and MHC loci. Associations are colored by gene-type instead of by gene identity for simplicity. Only SNP associations whose $P < 5 \times 10^{-6}$ are shown here. The gray horizontal line corresponds to a Bonferroni-corrected p-value significance threshold of $5.09 \times 10^{-11}$.

The online version of this article includes the following source data and figure supplement(s) for figure 1:

**Source data 1.** There are 9152 significant associations between the frequency of various V-, D-, and J-genes and the genotype of SNPs genome-wide.

**Source data 2.** Genomic inflation factor values are less than 1.03 for all paired gene-frequency, productivity GWAS analyses.

**Figure supplement 1.** For the significantly associated *TCRB* locus SNPs, the median association effect magnitude was largest for the expression of TRBD1 followed by the expression of TRBD2 and the expression of TRBV28 all in productive TCRs.

**Figure supplement 2.** For the significantly associated MHC locus SNPs, the median association effect magnitude was largest for the expression of TRBV4-1 followed by the expression of TRBV10-3.

**Figure supplement 3.** The majority of significantly associated *TCRB* locus SNPs had similar gene usage association P-values between non-productive and productive TCRs, but significantly associated MHC locus SNPs were only significant for gene usage of productive TCRs.

(*Oltz, 2001*), this difference in repression could subsequently change the usage frequency of the *TRBV24* gene.

## *DCLRE1C* locus variation is associated with the extent of V-, D-, and J-gene trimming

We hypothesized that SNPs across the genome, particularly those within V(D)J-recombination-associated genes, may influence the extent of TCR nucleotide trimming at V(D)J *TCRB* gene junctions.

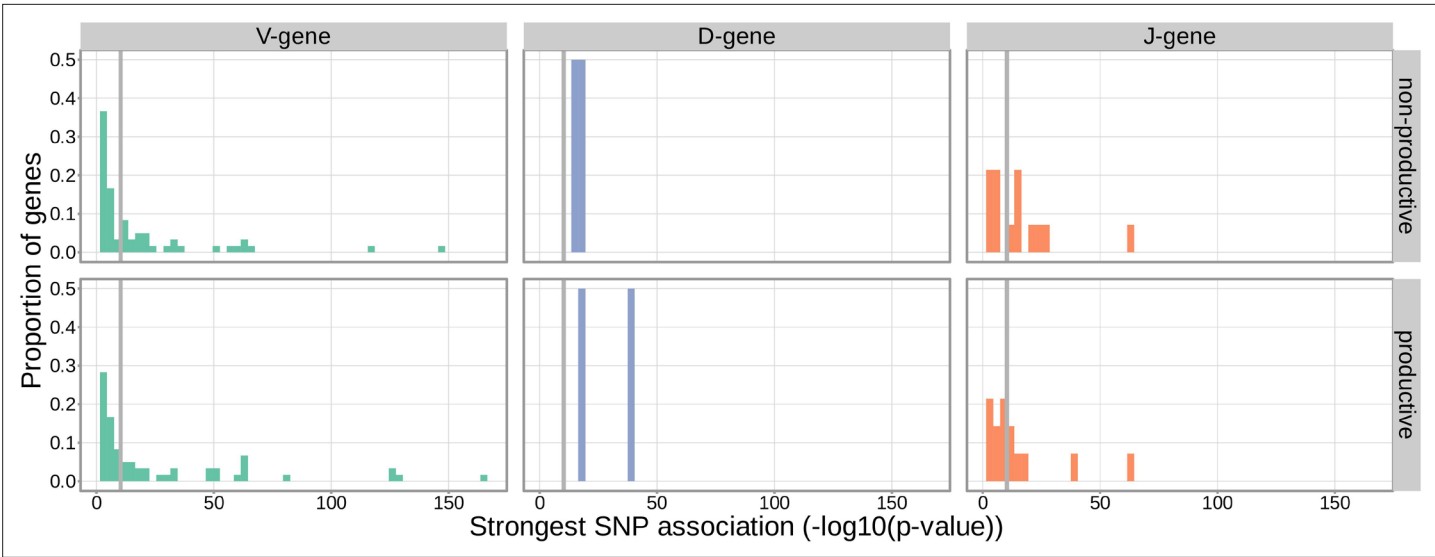

**Figure 2.** Gene-usage frequency of many V-gene, D-gene, and J-gene segments is significantly associated with variation in the *TCRB* locus. The p-value of the strongest *TCRB* SNP, gene-usage association for each different V-gene, D-gene, and J-gene segment is given on the X-axis. The proportion of gene segments within each gene type is given on the Y-axis. The gray vertical lines correspond to a whole-genome-level Bonferroni-corrected p-value significance threshold of $5.09 \times 10^{-11}$.

The online version of this article includes the following source data for figure 2:

**Source data 1.** Top *TCRB* SNP, gene-usage association p-value for each different V-gene, D-gene, and J-gene.

It has been previously observed that the extent of trimming varies by V(D)J *TCRB* gene choice (*Figure 3—figure supplement 4*; *Nadel and Feeney, 1995*; *Nadel and Feeney, 1997*; *Jackson et al., 2004*; *Murugan et al., 2012*). In other words, two different V-genes (*TRBV19* and *TRBV20-1*, for example) will on average be trimmed to different extents due, in part, to differences in their terminal nucleotide sequences (and the same is true for D- and J-genes). Thus, to quantify the effect of SNPs on the extent of V-, D-, and J-gene trimming during V(D)J recombination, without confounding the extent of trimming with *TCRB* gene choice, we designed a linear fixed effects model to measure the correlation between each SNP and the number of nucleotide deletions, while conditioning out the effect mediated by gene choice. We fit this 'gene-conditioned model' for each of the four trimming types (V-gene trimming, 5' end D-gene trimming, 3' end D-gene trimming, and J-gene trimming) on our paired data set. We performed the analysis, as above, incorporating ancestry-informative principal components in each model (detailed in Materials and methods). Diagnostic statistics show that this correction for population-substructure-related biases is sufficient (*Figure 3—source data 2*). Here, we considered the significance of associations at a Bonferroni-corrected whole-genome p-value significance threshold of $9.68 \times 10^{-10}$ (see Materials and methods).

With these methods, we identified 317 significant SNP associations with the extent of nucleotide trimming for various trimming types (*Figure 3* and *Figure 4—source data 1*). We found 66 highly significant associations between V- and J-gene trimming and SNPs within the *DCLRE1C* gene locus for both productive and non-productive TCRs when considered in the whole-genome context. For these significant *DCLRE1C* locus SNP associations, the magnitudes of the effects were greater for non-productive TCRs compared to productive TCRs for both V-gene trimming and J-gene trimming (*Figure 4—figure supplement 1*). The *DCLRE1C* gene encodes the Artemis protein, an endonuclease responsible for cutting the hairpin intermediate prior to nucleotide trimming and insertion during V(D)J recombination. Many of the SNPs responsible for these 66 significant associations within the *DCLRE1C* locus were shared between trimming and productivity types (*Figure 4*). The most significantly-associated SNP (rs41298872) within this locus had a p-value of $3.18 \times 10^{-37}$ for J-gene trimming of non-productive TCRs (*Figure 3—figure supplement 2*). This SNP was also significantly-associated with J-gene trimming of productive ($P = 1.99 \times 10^{-29}$) TCRs and V-gene trimming of productive ($P = 6.23 \times 10^{-23}$) and non-productive ($P = 2.81 \times 10^{-21}$) TCRs. We performed a conditional analysis to identify potential independent secondary signals by including this SNP as an additional covariate

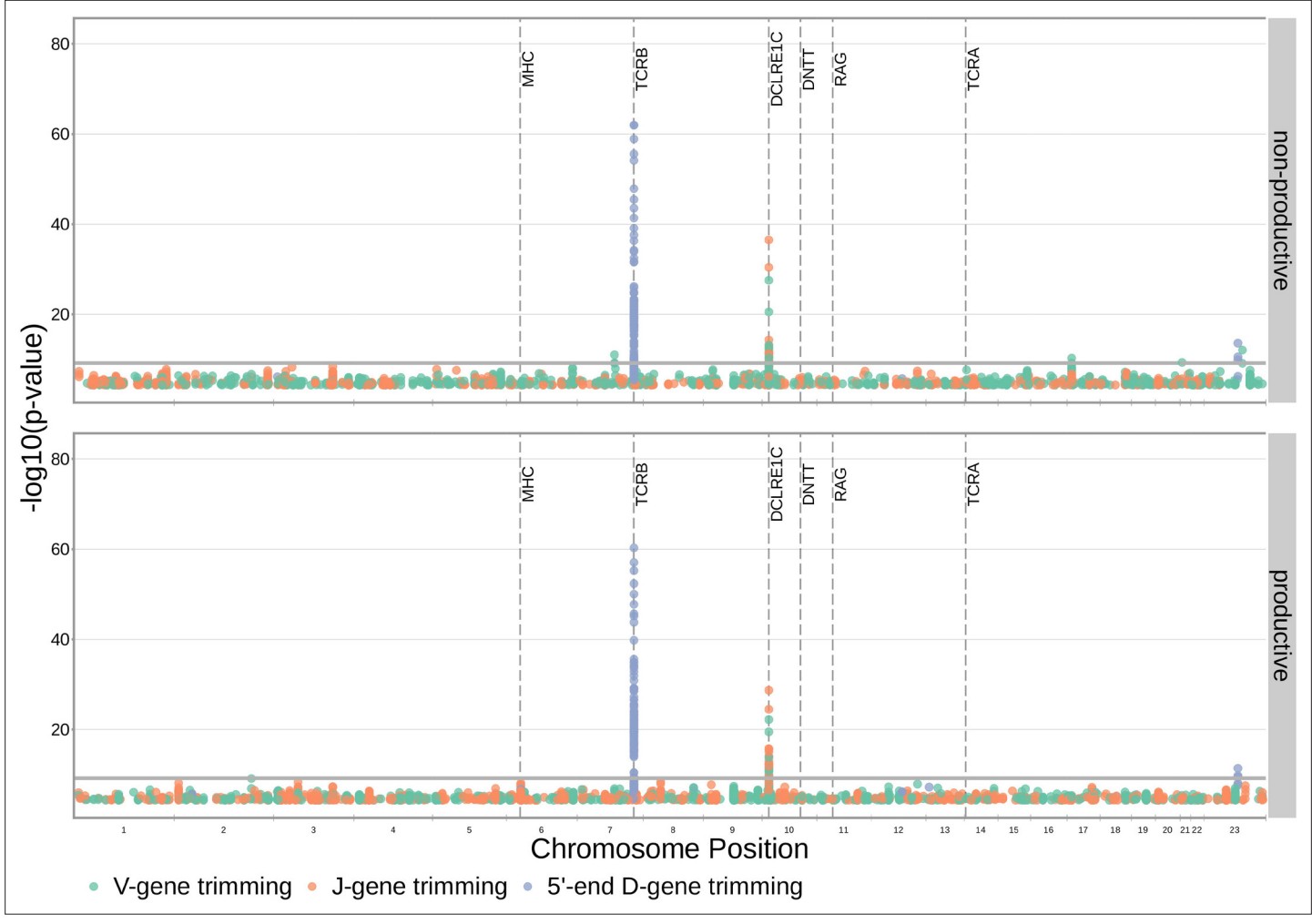

**Figure 3.** SNP associations for all four trimming types reveal the most significant associations to be located within the *TCRB* and *DCLRE1C* loci for 5′ D-gene trimming and J-gene trimming, respectively, when conditioning out effects mediated by gene choice when calculating the strength of association. Only SNP associations whose $P < 5 \times 10^{-5}$ are shown here. The gray horizontal line corresponds to a Bonferroni-corrected p-value significance threshold of $9.68 \times 10^{-10}$.

The online version of this article includes the following source data and figure supplement(s) for figure 3:

**Source data 1.** There are 317 significant SNP associations with the extent of nucleotide trimming for various trimming types.

**Source data 2.** Genomic inflation factor values are less than 1.03 for all paired nucleotide trimming, productivity GWAS analyses.

**Figure supplement 1.** The SNP genotype for the SNP (rs2367486) most significantly associated with 5′ end D-gene trimming within the *TCRB* locus is also associated with *TRBD2\*02* allele genotype.

**Figure supplement 2.** Significant associations are no longer observed between 5′ end D-gene trimming and variation in the *TCRB* locus after correcting for *TRBD2* allele genotype in our model formulation.

**Figure supplement 3.** Significant associations are also no longer observed between 5′ end D-gene trimming and variation in the *TCRB* locus when restricting the analysis to TCRs which contain *TRBJ1* genes (and consequently contain *TRBD1*).

**Figure supplement 4.** The extent of nucleotide deletion varies by the gene allele identity for all gene types.

**Figure supplement 5.** Significant SNP associations are located within the MHC, *TCRB*, and *DCLRE1C* loci for all four trimming types when calculating the strength of association without conditioning out effects mediated by gene choice.

**Figure supplement 6.** SNP associations for all fractions of non-gene-trimmed TCRs containing P-nucleotides are not significant within the *DCLRE1C* locus.

**Figure supplement 7.** SNP associations for the number of P-nucleotides are not significant within the *DCLRE1C* locus.

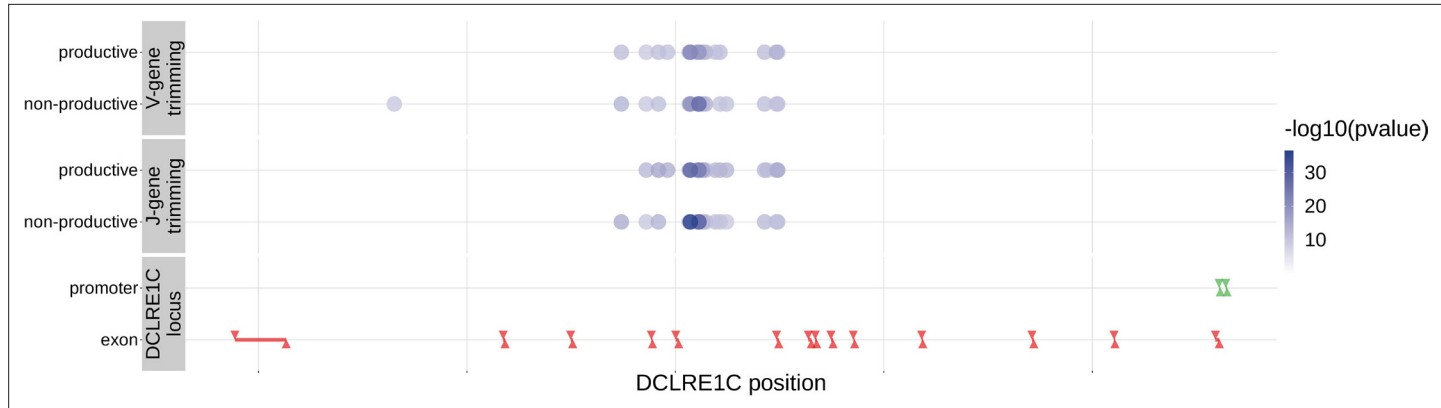

**Figure 4.** Within the *DCLRE1C* locus, 93.8% of these significantly associated SNPs were located within introns. Additionally, many of these significant SNP associations overlapped between trimming types. Downward arrows represent promoter/exon starting positions and upward arrows represent promoter/exon ending positions.

The online version of this article includes the following source data and figure supplement(s) for figure 4:

**Source data 1.** *DCLRE1C* locus SNP association p-values and locus positions.

**Source data 2.** There are two independent SNP signals within the *DCLRE1C* locus for J-gene trimming of non-productive TCRs.

**Figure supplement 1.** For the significantly associated *DCLRE1C* locus SNPs, the magnitudes of the effects were greater for non-productive TCRs compared to productive TCRs for both V-gene trimming and J-gene trimming.

**Figure supplement 2.** The extent of J-gene trimming changes as a function of SNP genotype for the SNP (rs41298872) most significantly associated with J-gene trimming within the *DCLRE1C* locus.

**Figure supplement 3.** The extent of V- and J-gene trimming of productive and non-productive TCRβ chains changes as a function of SNP genotype within the discovery cohort for a non-synonymous *DCLRE1C* SNP (rs12768894, c.728A>G).

**Figure supplement 4.** The extent of V-gene trimming.

**Figure supplement 5.** The extent of V- (**A**) and J-gene (**B**) trimming of productive and non-productive TCRα chains changes as a function of SNP genotype within the validation cohort for a non-synonymous *DCLRE1C* SNP (rs12768894, c728A>G).

within the model. This analysis revealed a second, independent SNP signal (rs35441642) within the *DCLRE1C* locus for J-gene trimming of non-productive TCRs (*Figure 4—source data 2*). None of the other nucleotide trimming type, productivity status combinations had significant evidence for secondary independent signals.

Our procedure also identified many highly significant associations between 5' end D-gene trimming and SNPs within the *TCRB* gene locus, however these appear to result from correlations between SNP genotype and *TRBD2* allele genotype (*Figure 3—figure supplement 1*). If we correct for *TRBD2* allele genotype in our model formulation (see Materials and methods), we no longer observe these associations between SNPs within the *TCRB* gene locus and the extent of 5' end D-gene trimming (*Figure 3—figure supplement 2*). *TRBD2* allele genotype could be acting as a confounding variable due to linked local genetic variation which influences nucleotide trimming and/or D-gene assignment ambiguity variation as a function of *TRBD2* allele genotype. To explore the extent of possible D-gene assignment ambiguity variation, we restricted our analysis to TCRs which contain *TRBJ1* genes and consequently contain *TRBD1* due to topological constraints during V(D)J recombination (*Robins et al., 2010*; *Murphy and Weaver, 2016*). With this approach, we also no longer observe associations between SNPs within the *TCRB* gene locus and the extent of 5' end D-gene trimming, and additionally, we do observe significant associations between SNPs within the *DCLRE1C* locus and 5' and 3' end D-gene trimming which were not observed in the original genome-wide analysis (*Figure 3—figure supplement 3*).

Our fixed effects model formulation for these inferences is important: if we don't condition on gene choice then additional, and presumably spurious, associations arise. Indeed, when implementing the 'simple model' designed to quantify the association between the four trimming types and genome-wide SNP genotypes, without conditioning out the effect mediated by gene choice, we observe additional associations between SNPs within the MHC locus and V-gene trimming of productive TCRs and between SNPs within the *TCRB* locus and V-gene and 3' end D-gene trimming of, again, productive

TCRs (*Figure 3—figure supplement 5*). This is perhaps not surprising, as we noted earlier that variations in the MHC and *TCRB* loci are associated with gene usage frequencies in productive TCRs (*Figure 1*), and different genes have different trimming distributions (determined in part by the nucleotide sequences at their termini).

Because P-nucleotides can be present at V(D)J junctions in the absence of nucleotide trimming (*Murphy and Weaver, 2016*), we hypothesized that similar *DCLRE1C* locus variation may also be associated with P-addition. Interestingly, we did not identify any strong associations between SNPs within the *DCLRE1C* locus and the fraction of non-gene-trimmed TCRs containing P-nucleotides when implementing our 'gene-conditioned model', despite the known role of the Artemis protein in functioning as an endonuclease responsible for cutting the hairpin intermediate, and thus, potentially creating P-nucleotides during V(D)J recombination (*Figure 3—figure supplement 6*). We observe similar results when quantifying the effect of genome-wide SNPs on the number of V-, D-, and J-gene P-nucleotides per TCR (*Figure 3—figure supplement 7*).

## *DNTT* locus variation is associated with the number of V-D and D-J N-insertions

Unlike V-, D-, or J-gene nucleotide trimming length, the number of nucleotide N-insertions between V-D and D-J genes does not vary substantially with V(D)J *TCRB* gene choice (*Figure 5—figure supplement 1*; *Murugan et al., 2012*). Thus, to infer the association between SNPs and the number of nucleotide N-insertions, we implemented a 'simple model', without conditioning out any effect mediated by gene choice. Again, because of the potential for population-substructure-related effects to inflate associations between each SNP and the number of N-insertions, we incorporated ancestry-informative principal components as covariates in each model (detailed in Materials and methods). Diagnostic statistics show that this bias correction is sufficient (*Figure 5—source data 3*).

With these methods, we identified three associations between SNPs and the number of nucleotide N-insertions using a Bonferroni-corrected whole-genome P-value significance threshold of $1.94 \times 10^{-9}$ (see Materials and methods) (*Figure 5* and *Figure 5—source data 1*). Two SNPs within the *DNTT* gene locus (rs2273892 and rs12569756) were responsible for these associations. The *DNTT* gene encodes the terminal deoxynucleotidyl transferase (TdT) protein which is a specialized DNA polymerase responsible for adding non-templated (N) nucleotides to coding junctions during V(D)J recombination. When we restrict our analysis to TCRs which contain *TRBJ1* genes and consequently eliminate potential D-gene assignment ambiguity, we continue to observe these *DNTT* associations (*Figure 5—figure supplement 2*).

Since the TdT protein has an important mechanistic role in the N-insertion process and because we already identified SNPs within the *DNTT* locus to be weakly associated with the number of N-insertions at V(D)J gene junctions, we wanted to explore the locus further. Restricting the analysis to the extended *DNTT* locus reduced the multiple testing burden such that 232 significant associations emerged (*Figure 5* and *Figure 5—source data 2*). For these significant *DNTT* locus SNP associations, the magnitudes of the effects were greater for non-productive TCRs compared to productive TCRs for both V-D-gene junction N-insertion and D-J-gene junction N-insertion (*Figure 6—figure supplement 1*). Many of the SNPs responsible for these 232 significant associations within the extended *DNTT* locus were shared between insertion and productivity types (*Figure 6*). While most of these associations are likely the result of a single independent signal for each insertion and productivity type, we performed a conditional analysis to identify potential independent secondary signals. To do so, we included the most significant SNP within the *DNTT* locus for each insertion and productivity type as a covariate in the model. With this approach, we identified rs2273892 as the primary independent signal for D-J N-insertion of non-productive TCRs and rs12569756 as the primary independent signal for D-J N-insertion of productive TCRs and V-D N-insertion of productive and non-productive TCRs. However, these two SNPs are tightly linked and, thus, likely both represent the same, primary independent signal. This analysis did not reveal any significant evidence for secondary independent signals.

We found that correcting for population-substructure-related effects was especially important in our primary genome-wide analysis, which led us to discover differences in the extent of N-insertion by ancestry-informative PCA cluster. Indeed, if we don't incorporate correction terms for population-substructure-related biases in our model formulation, we observe many strongly significant

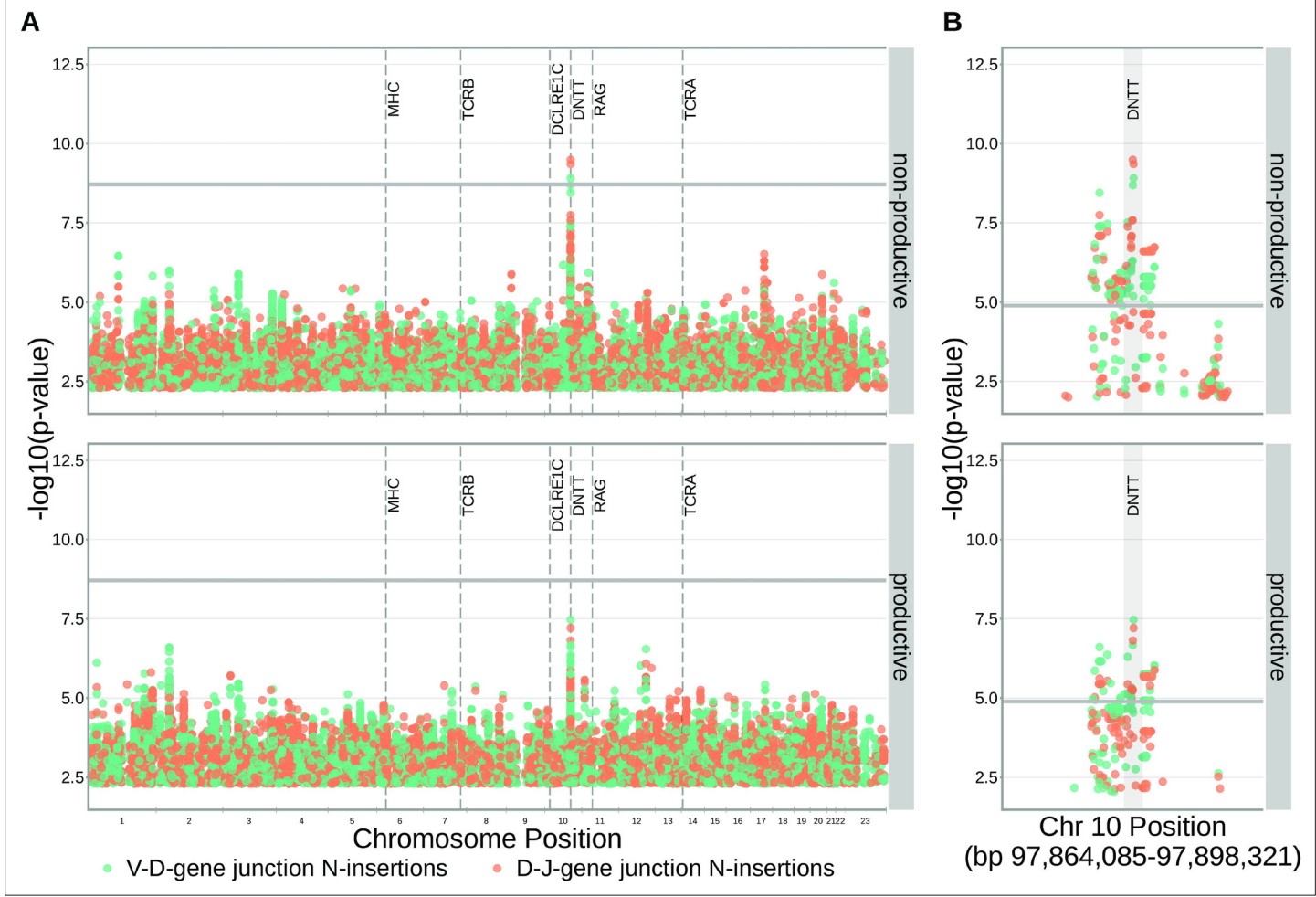

**Figure 5.** SNPs within the *DNTT* locus are associated with the extent of N-insertion. (**A**) There are three associations for SNPs within the *DNTT* locus which are significant when considered in the whole-genome context. The gray horizontal line corresponds to a whole-genome Bonferroni-corrected P-value significance threshold of $1.94 \times 10^{-9}$. (**B**) Using a *DNTT* gene-level significance threshold, many more SNPs within the extended *DNTT* locus have significant associations for both N-insertion types. Here, the gray horizontal line corresponds to a gene-level Bonferroni-corrected P-value significance threshold of $1.28 \times 10^{-5}$ (calculated using gene-level Bonferroni correction for the 977 SNPs within 200 kb of the *DNTT* locus, see Materials and methods). For both (**A**) and (**B**), only SNP associations whose $P < 5 \times 10^{-3}$ are shown.

The online version of this article includes the following source data and figure supplement(s) for figure 5:

**Source data 1.** There are three significant associations between SNPs genome-wide and the number of nucleotide N-insertions.

**Source data 2.** There are 232 significant associations between SNPs genome-wide and the number of nucleotide N-insertions when restricting the analysis to the extended *DNTT* locus.

**Source data 3.** Genomic inflation factor values are less than 1.03 for all paired N-insertion, productivity GWAS analyses.

**Figure supplement 1.** The extent of N-insertion does not vary substantially by the gene allele identity for any gene type.

**Figure supplement 2.** Significant associations continue to be observed within the *DNTT* locus for both V-D- and D-J-gene-junction N-insertions when restricting the analysis to TCRs which contain *TRBJ1* genes and consequently contain *TRBD1*.

associations, particularly within the *DNTT* locus. This hinted at important PCA-cluster level effects. When we look closely at the average number of N-insertions (combining the number of V-D and D-J N-insertions) across TCR repertoires by PCA cluster, we note that subjects from the 'Asian'-associated PCA cluster have significantly fewer total N-insertions for productive ($P = 0.006$ without Bonferroni correction) and non-productive ($P = 0.014$ without Bonferroni correction) TCRs when compared to the population mean (using a one-sample t-test) (*Figure 7*). The total N-insertions for productive TCRs within the 'Asian'-associated PCA cluster remain significantly different from the population mean after Bonferroni multiple testing correction (corrected $P = 0.036$). Furthermore, the 'Asian'- and

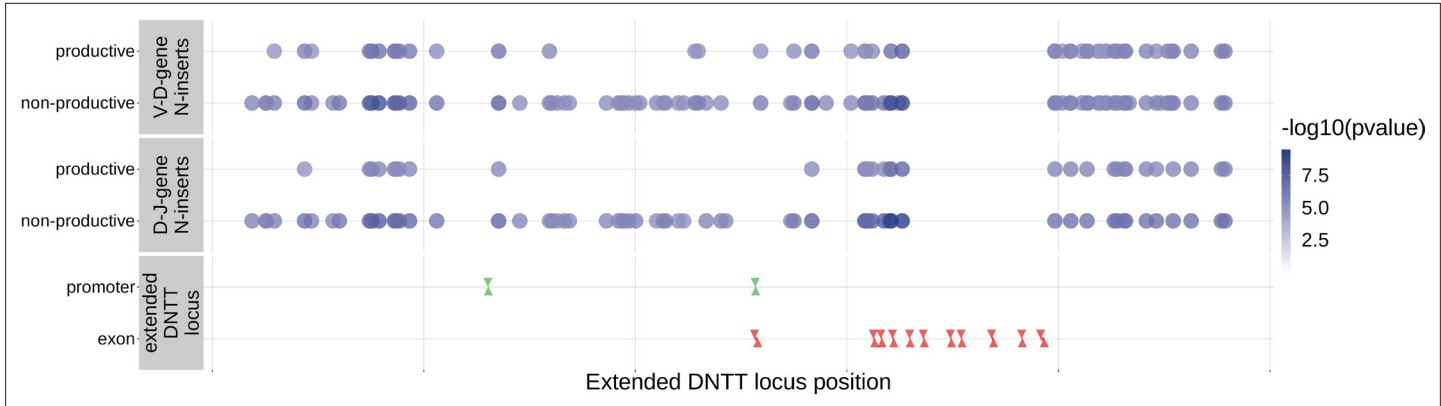

**Figure 6.** Within the *DNTT* locus, many of the significant SNP associations overlapped between N-insertion types when using *DNTT* gene-level Bonferroni-corrected p-value significance threshold of $1.28 \times 10^{-5}$. Downward arrows represent promoter/exon starting positions and upward arrows represent promoter/exon ending positions.

The online version of this article includes the following source data and figure supplement(s) for figure 6:

**Source data 1.** *DNTT* locus SNP association p-values and locus positions.

**Figure supplement 1.** For these significant *DNTT* locus SNP associations, the magnitudes of the effects were greater for non-productive TCRs compared to productive TCRs for both V-D-gene junction N-insertion and D-J-gene junction N-insertion.

**Figure supplement 2.** The extent of V-D and D-J N-insertion of productive and non-productive TCRβ chains changes as a function of SNP genotype within the discovery cohort for an intronic *DNTT* SNP (rs3762093).

**Figure supplement 3.** An intronic SNP (rs3762093) within the *DNTT* gene locus is not strongly associated with the number of V-D (**A**) or D-J (**B**) N-inserts within productive or non-productive TCRβ chains in the validation cohort.

**Figure supplement 4.** An intronic SNP (rs3762093) within the *DNTT* gene locus is significantly associated with the number of V-J N-inserts for productive TCRα chains in the validation cohort.

'Hispanic'-associated PCA clusters had significantly higher mean SNP allele frequencies for SNPs within the extended *DNTT* region that were associated with fewer N-insertions when compared to the mean population allele frequency ($P = 7.32 \times 10^{-20}$ for the 'Asian'-associated PCA cluster and $P = 1.17 \times 10^{-5}$ for the 'Hispanic'-associated PCA cluster using a one-sample t-test with Bonferroni multiple testing correction) (*Figure 8*).

## Validation analysis

To validate our results, we worked with paired ancestry-informative marker (AIM) SNP array and TCRα- and TCRβ-immunosequencing data representing 94 individuals and 2 SNPs (which overlap with the discovery dataset) from an independent validation cohort (*Table 3* and see Materials and methods). In contrast to the discovery cohort, this cohort contains different demographics, shallower RNA-seq-based TCR-sequencing, and a sparser set of SNPs. However, TCR-sequencing for both TCRα and TCRβ chains is available.

We were able to validate a discovery-cohort significantly associated *DCLRE1C* SNP within this validation cohort. While none of the independent *DCLRE1C* SNPs from the discovery-cohort analysis overlapped with the validation cohort SNP set, a single, non-synonymous SNP (rs12768894, c.728A > G) within the *DCLRE1C* locus was present in both SNP sets. This SNP was one of the significant associations we observed for V-gene trimming (productive $P = 2.16 \times 10^{-14}$; non-productive $P = 7.21 \times 10^{-14}$) and J-gene trimming (productive $P = 1.23 \times 10^{-11}$; non-productive $P = 6.62 \times 10^{-12}$) of TCRβ chains in the genome-wide discovery cohort analysis (*Figure 4—figure supplement 3*). Using the same methods, we identified significant associations between this SNP and J-gene trimming of productive TCRα and TCRβ chains and V-gene trimming of both productive and non-productive TCRα and TCRβ chains within the validation cohort (*Table 4*, *Figure 4—figure supplement 4*, and *Figure 4—figure supplement 5*). Associations between rs12768894 and both types of D-gene trimming of TCRβ chains were not significant for either cohort.

We were unable to validate the most significantly associated *DNTT* SNPs due to lack of overlap between the SNP sets for the discovery and validation cohorts; a discovery-cohort weakly associated

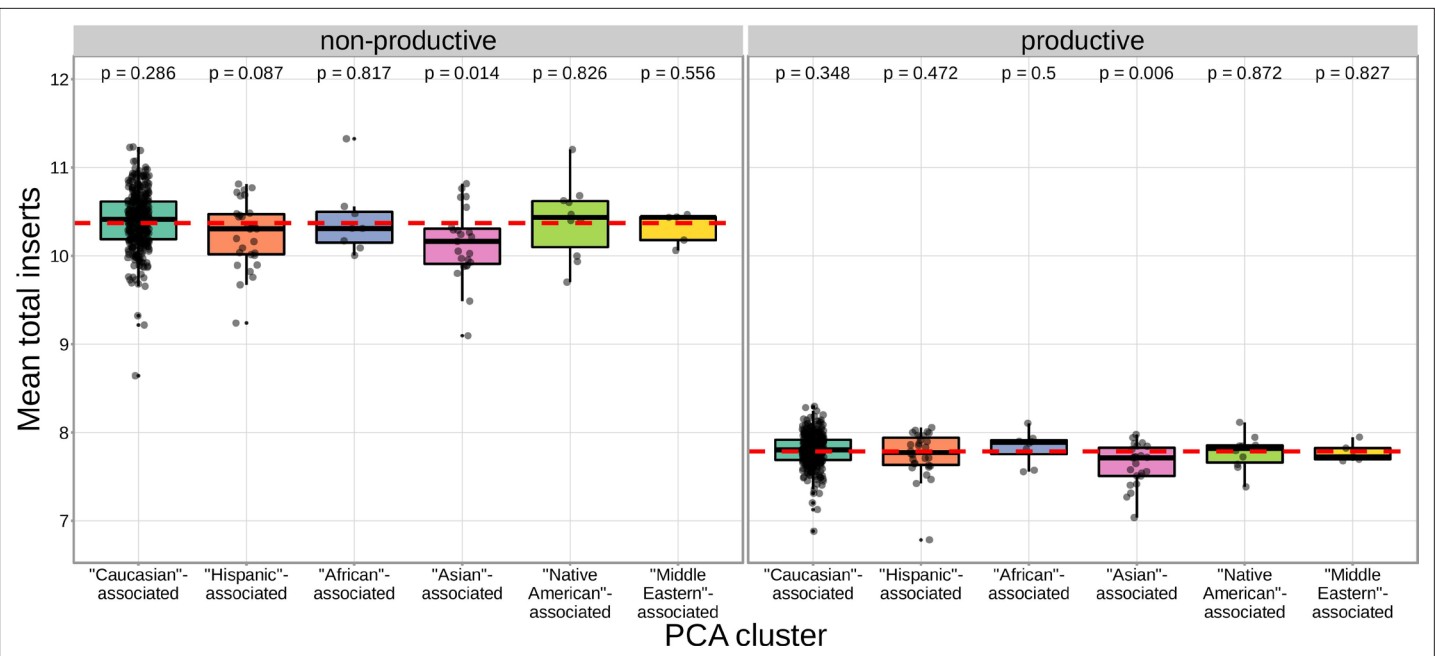

**Figure 7.** The TCR repertoires for subjects in the 'Asian'-associated PCA-cluster contain fewer N-insertions for productive TCRs when compared to the population mean computed across all 666 subjects (dashed, red horizontal line). The p-values from a one-sample t-test (without Bonferroni multiple testing correction) for each PCA cluster compared to the population mean are reported at the top of the plot.

The online version of this article includes the following source data and figure supplement(s) for figure 7:

**Source data 1.** PCA-cluster and average number of N-insertions by subject.

**Figure supplement 1.** The population mean is dominated by subjects in the 'Caucasian'-associated PCA-cluster.

SNP (rs3762093) failed to reach statistical significance for all N-insertion types, but had the same direction of effect in the validation cohort as follows. Within the discovery cohort, rs3762093 genotype was weakly associated with the number of V-D N-insertions (productive $P = 1.37 \times 10^{-6}$; non-productive $P = 1.50 \times 10^{-7}$) and D-J N-insertions (productive $P = 9.43 \times 10^{-6}$; non-productive $P = 1.94 \times 10^{-7}$) within TCRβ chains (*Figure 6—figure supplement 2*). Within the validation cohort, this SNP was significantly associated with the number of V-J N-insertions within productive TCRα chains (*Table 4* and *Figure 6—figure supplement 4*). However, this SNP was not significantly associated with the number of V-D or D-J N-insertions within productive or non-productive TCRβ chains or the number of V-J N-insertions within non-productive TCRα chains within the validation cohort (*Table 4*, *Figure 6—figure supplement 3*, and *Figure 6—figure supplement 4*). Despite the lack of significance, we noted that the model coefficients for rs3762093 genotype were in the same direction (i.e. the minor allele was associated with fewer N-insertions) for all N-insertion and productivity types within TCRβ chains for both cohorts. Further, while TCRα chain sequencing was not available for the discovery cohort, we observed stronger associations between rs3762093 genotype and the extent of N-insertion for both productivity types within TCRα chains compared to TCRβ chains within the validation cohort. Perhaps with a larger validation cohort, significant associations would be present for all N-insertion types.

## Discussion

V(D)J recombination is a complex stochastic process that enables the generation of diverse TCR repertoires. Our results show that genetic variation in various V(D)J recombination genes has a key role in shaping the TCR repertoire through biasing V(D)J gene choice, nucleotide trimming, and N-insertion in a broad population sample. While we recognize that there may be a complicated entanglement between allelic variation and local *cis*-acting effects, we were primarily interested in identifying strong, *trans*-acting associations. By leveraging the unique pairing of TCRβ chain immunosequencing and genome-wide genotype data, we have (1) confirmed and extended previous studies on the genetic determinants of TCR V-gene usage, (2) discovered associations between common genetic variants

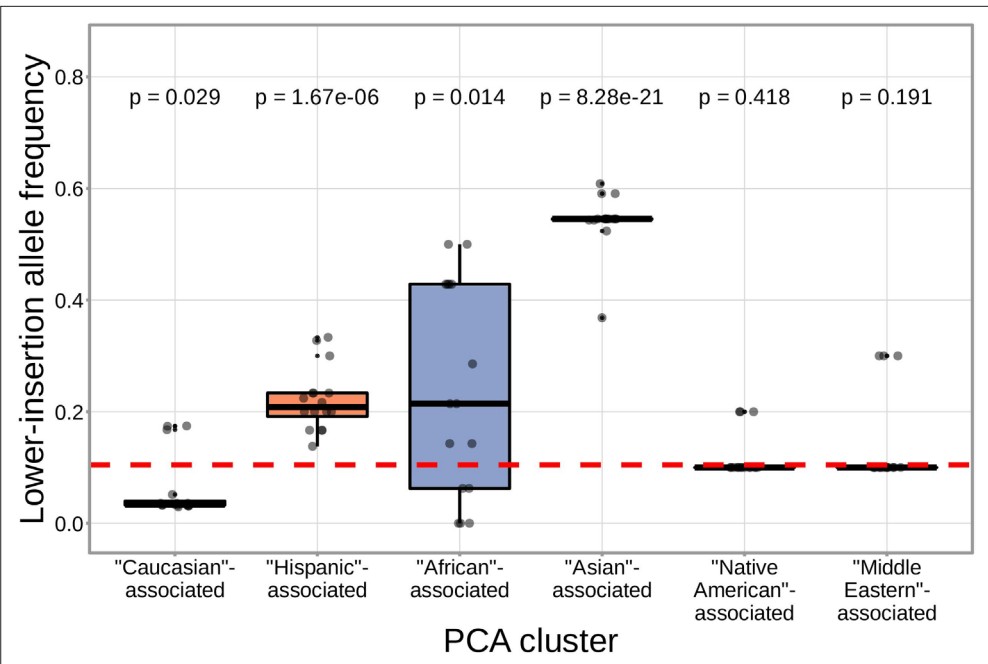

**Figure 8.** SNPs within the *DNTT* region that are associated with fewer N-insertions have a higher mean allele frequency within the 'Asian'-associated PCA-cluster when compared to the population mean allele frequency computed across the 398 discovery cohort subjects (dashed, red horizontal line). The p-values from a one-sample t-test (without Bonferroni multiple testing correction) for each PCA cluster compared to the population mean are reported at the top of the plot. The population mean is dominated by subjects in the 'Caucasian'-associated PCA cluster (*Figure 7—figure supplement 1*).

The online version of this article includes the following source data for figure 8:

**Source data 1.** Allele frequencies by PCA-cluster for SNPs within the *DNTT* locus that are associated with fewer N-insertions.

---

within the *DCLRE1C* and *DNTT* loci and V(D)J junctional trimming and N-insertions, respectively, (3) developed a method for quantifying the extent of the associations between genetic variations and junctional features, directly, without confounding gene choice effects, and (4) revealed differences in the extent of N-insertion by ancestry-informative PCA cluster.

We note an abundance of associations between variation in the *TCRB* locus and V(D)J gene usage biases for both productive and non-productive TCRs. Although previous reports have revealed similar patterns of association for productive TCRs (*Sharon et al., 2016*; *Gao et al., 2019*), our results refine and extend this result by quantifying the extent of *TCRB* locus variation on V(D)J gene usage for non-productive TCRs. This highlights that locus variation is associated with TCR generation-related gene usage biases, in addition to potential thymic selection biases for productive TCRs. These TCR generation-related gene usage biases likely reflect local gene regulation and/or recombination efficiency effects. For example, one of the SNPs most significantly associated with *TRBV28* expression (rs17213) is located within the recombination signal sequence at the 3'-end of the gene and, thus, could be involved directly

**Table 3.** Validation cohort demographics.

| | | Count |
|---|---|---|
| **Sex** | Female | 58 |
| | Male | 36 |
| **Age (in years)** | < 10 | 26 |
| | 11–20 | 15 |
| | 21–30 | 13 |
| | 31–40 | 12 |
| | 41–50 | 11 |
| | 51–60 | 9 |
| | > 60 | 8 |
| **Self-reported ethnicity** | Hispanic or Latino | 94 |
| **CMV serostatus** | Positive | 37 |
| | Negative | 57 |
| **Total** | | 94 |

**Table 4.** We inferred the associations between SNP genotype and TCR repertoire features for two SNPs overlapping between discovery-cohort and validation-cohort SNP sets.

We considered the significance of the validation cohort associations at a Bonferroni-corrected SNP-level p-value significance threshold of 0.0042 for trimming and 0.0083 for N-insertion (see Materials and methods). Validation cohort p-values are one-tailed. * Discovery-cohort associations were only significant when considered at the *DNTT* -gene level significance threshold, not at the whole-genome significance threshold.

| SNP | TCR chain | Repertoire feature | Productivity type | Discovery cohort significant association | Validation cohort significant association |
|---|---|---|---|---|---|
| **rs12768894** | TCRβ | V-gene trimming | Productive | Yes ($2.16 \times 10^{-14}$) | Yes ($7.17 \times 10^{-7}$) |
| | | | Non-productive | Yes ($7.21 \times 10^{-14}$) | Yes ($8.75 \times 10^{-6}$) |
| | | J-gene trimming | Productive | Yes ($1.23 \times 10^{-11}$) | Yes ($5.16 \times 10^{-10}$) |
| | | | Non-productive | Yes ($6.62 \times 10^{-12}$) | No ($4.18 \times 10^{-2}$) |
| | TCRα | V-gene trimming | Productive | N/A | Yes ($2.59 \times 10^{-5}$) |
| | | | Non-productive | N/A | Yes ($2.68 \times 10^{-7}$) |
| | | J-gene trimming | Productive | N/A | Yes ($6.29 \times 10^{-12}$) |
| | | | Non-productive | N/A | No ($9.99 \times 10^{-3}$) |
| **rs3762093** | TCRβ | V-D N-insertion | Productive | Yes* ($1.37 \times 10^{-6}$) | No (0.153) |
| | | | Non-productive | Yes* ($1.50 \times 10^{-7}$) | No (0.059) |
| | | D-J N-insertion | Productive | Yes* ($9.43 \times 10^{-6}$) | No (0.137) |
| | | | Non-productive | Yes* ($1.94 \times 10^{-7}$) | No (0.006) |
| | TCRα | V-J N-insertion | Productive | N/A | Yes (0.006) |
| | | | Non-productive | N/A | No (0.031) |

in changing the recombination efficiency of *TRBV28*. Thus, different expression levels of various genes could be promoted by variation within non-coding regions such as promoters, 5'UTRs and leader sequences, introns, or recombination signal sequences. Polymorphisms within these regions have been suggested to influence V(D)J gene expression levels within B-cell receptor repertoires (*Mikoczlova et al., 2021*). We also observed that variation in the MHC locus is associated with V-gene usage biases for productive TCRs, but not non-productive TCRs. These MHC locus associations are likely only observed for V-gene usage since the V-gene locus, exclusively, encodes the TCR regions (complementarity-determining regions 1 and 2) which directly contact MHC during peptide presentation (*Murphy and Weaver, 2016*). While significant associations between MHC locus variation and V-gene usage have been identified previously (*Sharon et al., 2016*; *Gao et al., 2019*), the specific MHC locus variants and V-genes responsible for the most significant of these associations differed between the two studies and from those reported here. This variation is likely the result of population composition and/or exposure history differences between the various study cohorts. Despite their differences, both previous studies have suggested that the thymic selection of certain V-genes may be biased by germline-encoded TCR-MHC compatibilities in an MHC dependent manner (*Sharon et al., 2016*; *Gao et al., 2019*). Because of our observed distinction between associations present between MHC variation and V-gene usage in productive versus non-productive TCRs, our work supports this hypothesis.

We have identified, for the first time, specific genetic variants which are associated with modifying the extent of N-insertion and nucleotide trimming. While many previous studies have reported evidence of genetic influences on overall gene usage (*Zvyagin et al., 2014*; *Qi et al., 2016*; *Rubelt et al., 2016*; *Pogorelyy et al., 2018*; *Tanno et al., 2020*; *Fischer et al., 2021*) and repertoire similarity in response to acute infection (*Qi et al., 2016*; *Pogorelyy et al., 2018*), there have been few explorations into how heritable factors may bias TCR junctional features beyond reports of genetic similarity implying overall TCR repertoire similarity (*Krishna et al., 2020*; *Rubelt et al., 2016*). Here, we noted that variation in the gene encoding the Artemis protein (*DCLRE1C*) is associated with the extent of V- and J-gene nucleotide trimming for both productive and non-productive TCRs. These associations are strongest for non-productive TCRs suggesting a TCR generation-related repertoire bias. It is well established that the Artemis protein, in complex with DNA-PKcs, functions as an endonuclease

responsible for cutting the hairpin intermediate, and thus, potentially creating P-nucleotides prior to nucleotide trimming during V(D)J recombination (*Weigert et al., 1978*; *Moshous et al., 2001*; *Ma et al., 2002*; *Lu et al., 2007*). The direct involvement of Artemis in the nucleotide trimming mechanism, however, has yet to be confirmed. It has been shown that the Artemis protein possesses single-strand-specific 5' to 3' exonuclease activity (*Ma et al., 2002*; *Li et al., 2014*) and, thus, may be properly positioned to trim nucleotides. A non-synonymous SNP within *DCLRE1C* (rs12768894, c.728A > G) was one of the significant associations we observed for V- and J-gene nucleotide trimming in both the primary cohort and the independent validation cohort. Perhaps this mutation, or other linked non-synonymous *DCLRE1C* variation that was not studied here, is directly involved in the trimming changes we observe. We did not observe strong associations between variation in the *DCLRE1C* locus and the number of P-nucleotides or the fraction of non-gene-trimmed TCRs containing P-nucleotides, despite the established mutually exclusive relationship between P-addition and nucleotide trimming (*Gauss and Lieber, 1996*; *Srivastava and Robins, 2012*; *Murphy and Weaver, 2016*). However, the absence of P-nucleotide associations at the *DCLRE1C* locus could be the result of restricting the analyses to the non-gene-trimmed repertoire subset. Perhaps with a larger dataset these associations would be present.

Further, we have identified associations between variation in the gene encoding the TdT protein (*DNTT*) and the number of N-insertions for both productive and non-productive TCRs. Because of the established, direct involvement of the TdT protein in the N-insertion mechanism, these *DNTT* locus variations could be influencing the function of the TdT protein. These significant associations were slightly stronger for non-productive TCRs perhaps suggesting that thymic selection may limit the mechanistic effects of locus variation. Interestingly, we noted that the extent of N-insertion varies by ancestry-informative PCA cluster. Specifically, we found that the 'Asian'-associated PCA cluster had significantly fewer N-insertions for productive TCRs when compared to the population mean which is dominated by the 'Caucasian'-associated PCA cluster. This finding is, perhaps, related to the influence of broad heritable factors biasing the extent of N-insertions.

The significant SNPs associated with changing the extent of nucleotide trimming and N-insertion identified here could be expression quantitative trait loci (eQTLs); however, experimental work will be required to determine whether these SNPs modify the expression of *DCLRE1C* and *DNTT*, respectively. More work is also required to elucidate the mechanistic relationship between *DCLRE1C* locus variation and nucleotide trimming changes. After characterizing these relationships, future work can focus on identifying correlations between TCR repertoires and host immune exposures while accounting for genetically determined repertoire biases identified here. These directions would allow us to continue disentangling the genetic and environmental determinants governing TCR repertoire diversity.

There are several key limitations of our approach which are intrinsic to the data used in this study. First, the lack of overlap between SNP sets for the discovery and validation cohorts limited our ability to directly validate our strongest inferences. Next, it is possible that the SNP array data used here does not capture all potential causal variation. As such, a significantly associated SNP present in our SNP array data could simply be in linkage disequilibrium with a causal SNP which was either poorly imputed or not tested here. Previous work has suggested that polymorphisms within the immunoglobulin V-gene region are not completely captured by existing SNP array technology, and have been underrepresented in previous genome-wide association studies (*Watson and Breden, 2012*). SNP coverage of the TCRβ locus is thought to be even sparser (*Omer et al., 2022*), and thus, much of the actual TCRβ variation present within our data cohort is likely not captured by the SNP dataset used here (which contains 7,304 SNPs within the TRB locus, hg19:chr7:141950000–142550000). Lastly, we have used the recombination statistics from non-productive rearrangements here as a means of studying the V(D)J recombination generation process in the absence of selection; however, we acknowledge that the repertoire of non-productive rearrangements may be an imperfect proxy for a pre-selection TCR repertoire. Since each non-productive rearrangement is sequenced due to the presence in the same T cell of a successful rearrangement that survived selection, it is possible that within-cell correlation between rearrangement events could imprint selection effects onto the non-productive repertoire. However, we are not aware of any evidence for a mechanism in which productive and non-productive recombination events at the TCRβ locus are significantly correlated. As such, we are assuming that the productive and non-productive recombination events are independent, and thus, the recombination statistics from the repertoire of non-productive rearrangements should reflect

that of a pre-selection repertoire as is common in the literature (*Robins et al., 2010*; *Murugan et al., 2012*; *Zvyagin et al., 2014*; *Rubelt et al., 2016*; *Pogorelyy et al., 2018*).

Another key constraint is the challenge of inferring the V(D)J rearrangements from the final nucleotide sequences due to the poor characterization of the *TCRA* and *TCRB* loci. The *TCRA* and *TCRB* regions have been historically difficult to reliably map using short read sequencing due to their repetitive and complex nature. While recent work has identified many new *TRBV* alleles, many more undocumented *TRBV* alleles likely remain to be discovered (*Omer et al., 2022*). As such, the incomplete characterization of the *TCRB* locus limited our ability to infer the correct V(D)J -gene allele for each final nucleotide sequence. Further, the TCR sequencing technology used here leverages relatively short-read sequencing which captures only a portion of the V-gene present in each sequence. Because many *TRBV* alleles are identical to other *TRBV* alleles for much of the V-gene region present in these sequences, it can be difficult to unambiguously assign V-gene usage to the final nucleotide sequences. D-gene usage assignment is also challenging due to the short length of the *TRBD* alleles (12–16 nucleotides before nucleotide trimming and N-insertion). We have found that controlling for D-gene assignment ambiguity in the nucleotide trimming and N-insertion analyses results in similar significant associations within the *DNTT* and *DCLRE1C* loci. Although we cannot rule out some effect of incorrect V(D)J -gene assignment bias for *trans* associations resulting from the signal being 'masked' by stronger *TCRB* locus signals, these biases seem to be mostly restricted to *cis* associations.

In summary, we have found that the usage of *TCRB* genes is associated with variation in MHC and *TCRB* loci, the number of N-insertions is associated with *DNTT* variation, and the extent of nucleotide trimming is associated with *DCLRE1C* variation. Our results clearly demonstrate how variation in V(D)J recombination-related genes can bias TCR repertoire combinatorial and junctional diversity. In the case of B cells, genetically determined V(D)J gene usage biases within B-cell receptor repertoires have been linked to functional consequences for the overall immune response to specific antigens and, thus, an increased susceptibility to certain diseases (*Mikocziova et al., 2021*). As such, the genetic TCR repertoire biases identified here lay the groundwork for further exploration into the diversity of immune responses and disease susceptibilities between individuals. Such studies will enhance our understanding of how an individual's diverse TCR repertoire can support a unique, robust immune response to disease and vaccination. Our findings also provide a step towards the ability to understand and predict an individual's TCR repertoire composition which will be critical for the future development of personalized therapeutic interventions and rational vaccine design.

# Materials and methods

## Key resources table

| Reagent type (species) or resource | Designation | Source or reference | Identifiers | Additional information |
|---|---|---|---|---|
| Software, Algorithm | TCRdist | *Dash et al., 2017*, *Bradley et al., 2017* | | Version 0.0.2; Software can be found on GitHub |
| Software, Algorithm | migec | *Shugay et al., 2014* | RRID: SCR_016337 | Version 1.2.9; Software can be found on GitHub |
| Antibody | CD3-PerCP eFluor710 (Mouse monoclonal) | Thermo Fisher Scientific | Cat: 46-0037-42; RRID: AB_1834395 | 0.012 µg per 1 million cells (1:100) |
| Antibody | CD4-BV650 (Mouse monoclonal) | BD Biosciences | Cat: 563875; RRID: AB_2687486 | 2 µl per 1 million cells (1:50) |
| Antibody | CD8-APC Fire750 (Mouse monoclonal) | Biolegend | Cat: 344746; RRID: AB_2572095 | 0.1 µg per 1 million cells (1:100) |
| Antibody | TCRγ/$\delta$-PE Cy7 (Mouse monoclonal) | Biolegend | Cat: 331222; RRID: AB_2562891 | 1 µg per 1 million cells (1:40) |
| Other | Fc Block | BD Biosciences | Cat: 564220; RRID: AB_2728082 | 2.5 µg per 1 million cells (1:20) |
| Other | Live/Dead Aqua | Tonbo Biosciences | Cat: 13–0870 T100 | 1 µl per 1 million cells (1:100) |
| Commercial assay, kit | Qiagen QIAamp DNA Mini Kit | Qiagen | Cat: 51,306 | |

*Continued on next page*

*Continued*

| Reagent type (species) or resource | Designation | Source or reference | Identifiers | Additional information |
|---|---|---|---|---|
| Commercial assay, kit | Taqman SNP Genotyping Assay | Thermo Fisher Scientific | Cat: 4351379 | |
| Commercial assay, kit | TaqMan Genotyping Master Mix | Thermo Fisher Scientific | Cat: 4371353 | |

## Discovery cohort dataset

TCRβ repertoire sequence data for 666 healthy bone marrow donor subjects was downloaded from the Adaptive Biotechnologies website using the link provided in the original publication (*Emerson et al., 2017*). For both the discovery and validation cohorts, V, D, and J genes were assigned by comparing the TCRβ-chain (and TCRα-chain for the validation cohort) nucleotide sequences to the human IMGT/GENE-DB *TCRB* (or *TCRA*) allele sequences (*Giudicelli et al., 2005*). To infer the extent of nucleotide trimming, N-insertion, and P-addition for each TCRβ-chain (and TCRα-chain) nucleotide sequence, the most parsimonious V(D)J recombination scenario was assigned to each sequence using the TCRdist pipeline (*Dash et al., 2017*). The V(D)J recombination scenario requiring the fewest N-insertions was defined as the most parsimonious scenario.

SNP array data corresponding to 398 of these subjects was downloaded from The database of Genotypes and Phenotypes (accession number: phs001918). Details of the SNP array dataset, genotype imputation, and quality control have been described previously (*Martin et al., 2020*).

## Validation cohort dataset

Peripheral blood mononuclear cell (PBMC) samples were collected from 150 healthy subjects recruited at the Health Center Sócrates Flores Vivas (HCSFV) in Managua, Nicaragua (*Ng et al., 2016*). Healthy participants were recruited as contacts of influenza infected index patients and blood samples were collected at both the initial visit and a 30-day follow-up visit. Participants provided written informed consent and parental permission was obtained from parents or legal guardians of children, in addition to verbal assent from children aged 6 years and older. This study was approved by the Institutional Review Boards at the University of Michigan (HUM 00091392) and the Centro Nacional de Diagnóstico y Referencia (Ministry of Health, Nicaragua; CIRE 06/07/10–025).

With these samples, PBMCs were stained with CD3-PerCP eFluor710 (Thermo Cat. 46-0037-42), CD4-BV650 (BD Biosciences Cat. 563875), CD8-APC Fire750 (Biolegend Cat. 344746), and gdCy7 (Biolegend Cat. 331222). Briefly, after thawing from cryopreservation and plating in a 96-well round bottom plate, cells were spun down and resuspended in 50 µL of human Fc block (BD Biosciences Cat. 564220) in Dulbecco's phosphate-buffered saline (DPBS) at 5 µL per test (one test = $1.0 \times 10^6$ cells) and incubated for 10 min at room temperature. Afterwards, 50 µL of a 2 X Live/Dead Aqua (Tonbo Cat. 13–0870 T100, 1 µL per test, 1 test = $1.0 \times 10^6$ cells) and pre-titrated surface antibody cocktail in DPBS were added to each well and cells were incubated for 30 min on ice and in the dark. Cells were washed, resuspended in sort buffer and bulk sorted into polystyrene tubes. Afterwards, samples were spun down, pellets were resuspended in 350 µL of RNA lysis buffer, and stored at –80°C in labeled epitubes.

From here, DNA was extracted from 200 µL of neutrophil pellets using the Qiagen QIAamp DNA Mini Kit (Cat. 51306). Bulk repertoires for sorted CD4 and CD8 T cells were generated in accordance with the protocol developed by *Egorov et al., 2015*, and sequencing was performed on the NovaSeq by the Hartwell Center at St. Jude. Raw cDNA sequencing data were processed with the MIGEC software package (*Shugay et al., 2014*) to define error-corrected TCRA and TCRB transcript sequences, which were then analyzed as described above for the discovery cohort data (*Emerson et al., 2017*).

Genotypes for SNPs of interest corresponding to 94 of these subjects were pulled from Infinium Global Screening Array-24 v3.0 BeadChip results, which measures 654,027 SNP markers including multi-ethnic genome-wide content, curated clinical research variants, and quality control markers. High quality DNA was extracted using the Qiagen QIAamp DNA Mini Kit (Cat. 51306), and submitted to the St. Jude Hartwell Center for preparation and processing. Two SNPs, rs72640001 and rs72772435, were not included on this chip and were determined using Thermo Fisher TaqMan SNP Genotyping

Assays (Cat. 4351379, Assay ID C__99271581_10 and C__99587751_10, respectively) and TaqMan Genotyping Master Mix (Cat. 4371353) according to the kit manual.

## Data preparation

With these paired SNP array and TCR-immunosequencing for both the discovery and validation cohorts, we aimed to identify significant associations between these SNPs and various TCR repertoire features. Because we would expect a difference in these phenotypes depending on whether a TCR sequence is classified as productive or non-productive, we split the data based on this TCR productivity status and computed associations separately for the two groups.

We also subset the SNP data further based on several quality control metrics. We filtered the SNP array data to use only SNPs with a minor allele frequency above 0.05 in our analyses which excluded SNPs for which all subjects had the same genotype. For the discovery cohort, this filtering procedure and previous quality control (*Martin et al., 2020*) left 6,456,824 SNPs (of the original 35 million SNPs) remaining for our analyses. Only 2 SNPs from the validation cohort overlapped with this discovery cohort SNP set. For each of these discovery and validation cohort SNPs, when fitting each association model, we excluded observations which contained a missing SNP genotype. Next, for the TCR repertoire data, we excluded repertoires which contained a relatively small number of TCRs ($\log_{10}(\text{TCR count}) < 4.25$ for productive TCRs and $\log_{10}(\text{TCR count}) < 3.5$ for non-productive TCRs) from the analyses. Also, when fitting models for gene usage (i.e. V-gene usage, D-gene usage, and J-gene usage) we have restricted our analyses to observations which contain non-orphan genes. Lastly, for TCRβ-chains, if a D-gene is trimmed so much that the D-gene is unidentifiable, the inference pipeline used to infer *TCRB* genes for each sequenced TCR does not report a D-gene. Instead, this D-gene (if it is indeed present) is reported as a V-J N-insertion. Because of this, we excluded these observations when fitting models for TCR features involving the D-gene (i.e. D-gene usage, both V-D and D-J junction N-insertions, D-gene P-additions, and D-gene nucleotide trimming).

## Notation

The discovery dataset contains observations for a total of $I = 398$ subjects and the validation dataset contains observations for a total of $I = 94$ subjects. Within each cohort, for subject $i \in \{1, \ldots, I\}$, we observe a total of $N_i$ TCRs which, here, represents the number of TCRs which compose each subject's TCR repertoire. Thus, for each TCR $k \in \{1, \ldots, N_i\}$, we measure a TCR feature of interest, $y_{ik}$, such as the number of V-D N-insertions, the extent of V-trimming, etc. We also have SNP genotype data for a total of $J$ SNPs such that for each SNP $j \in \{1, \ldots, J\}$ and subject $i \in \{1, \ldots, I\}$, we measure the number of minor alleles in the genotype, $x_{ij} \in \{0, 1, 2\}$.

## Quantifying the association strength between each SNP and TCR feature using the 'simple model'

We first describe what we call the 'simple model'. We will describe more complex models, as well as each model with added correction for population-substructure-related effects, in the sections following.

We began by calculating the average occurrence of the TCR feature of interest, $\bar{y}_i$, within the repertoire of each subject, $i$. By condensing the data in this way, for each subject $i \in \{1, \ldots, I\}$, we are left with $N_i = 1$ observations. For example, for the discovery cohort, we can fit the model across $\sum_{i=1}^{I} N_i = 398$ observations. Using this condensed dataset, for each SNP, TCR feature, and productivity status, we can fit the model:

$$\bar{y}_i = x_{ij} \cdot \beta_{1j} + \beta_0 + \epsilon_{ij} \tag{1}$$

where $\beta_{1j}$ is the allele effect for SNP $j$ on the TCR feature of interest $\bar{y}_i$, $\beta_0$ is the intercept, and $\epsilon_{ij}$ is the random error for subject $i$ and SNP $j$ such that $\epsilon_{ij} \sim \mathcal{N}(0, \sigma^2)$.

To estimate each regression coefficient, we solved the least squares problem:

$$(\hat{\beta}_0, \hat{\beta}_{1j}) = \text{argmin}_{\beta_0, \beta_{1j}} \sum_{i=1}^{n} \left( \bar{y}_i - (x_{ij} \cdot \beta_{1j} + \beta_0) \right)^2 \tag{2}$$

using the function `lm` in R. With each estimate of the $j$-th SNP effect on the TCR feature of interest, $\hat{\beta}_{1j}$, generated by fitting the least squares problem (*Equation 2*), we quantified the association strength

between each SNP and the TCR feature of interest by testing whether $\hat{\beta}_{1j} = 0$. To do this, we calculate the test statistic

$$T_j = \frac{\hat{\beta}_{1j}}{\text{se}(\hat{\beta}_{1j})} \tag{3}$$

and compare $T_j$ to a $N(0,1)$ distribution to obtain each P-value.

## Quantifying the association strength between each SNP and TCR feature, conditional on *TCRB* gene type using the 'gene-conditioned model'

We noted that the amount of certain TCR features (such as the extent of all types of nucleotide trimming) vary by V(D)J *TCRB* gene choice. Thus, we can condition on this gene choice to quantify the direct association between each SNP and the amount of each TCR feature, without confounding gene choice effects. In this way, we condition on each gene type $t \in \{\text{V-gene, J-gene, D-gene}\}$ corresponding to the TCR feature of interest (i.e. $t = \text{V-gene}$ for V-gene trimming, $t = \text{J-gene}$ for J-gene trimming, etc.). We will refer to the following model as the 'gene-conditioned model' in the main text. Many similarities exist between the 'simple model' described in the previous section and this 'gene-conditioned model'. Thus, we will focus on the differences between the two models here. We will describe both models with added correction for population-substructure-related effects, in the sections following.

As in the previous section, we, again, want to reduce the number of data observations. For each subject $i \in \{1, \ldots, I\}$, we can calculate the average amount of each TCR feature $\bar{y}_{im}$ by each candidate *TCRB* gene allele group $m$ for the given gene type $t$ such that $m \in \{1, \ldots, M_t\}$. In calculating the average amount of each TCR feature across TCRs with the same candidate *TCRB* gene allele, we combined *TCRB* gene alleles which had identical CDR3 sequences and were of the same candidate *TCRB* gene into *TCRB* gene allele groups. As such, the number of observations per subject $N_i$ in this condensed dataset will equal $M_t$ and, thus, we will need to fit each model across $\sum_{i=1}^{I} M_t$ observations. In our data, for TCRβ chains, we observe 141 possible *TCRB* V-gene allele groups, 16 J-gene allele groups, and 3 D-gene allele groups. Thus, using the extent of nucleotide trimming as an example TCR feature within the discovery cohort, with this condensed formulation, for each SNP and productivity status, we have $\sim 56{,}000$ observations for V-gene trimming, $\sim 6{,}000$ observations for J-gene trimming, and $\sim 1{,}200$ observations for both types of D-gene trimming.

Using this condensed dataset, for each SNP, TCR feature, and productivity status, we fit the following 'gene-conditioned model':

$$\bar{y}_{im} = x_{ij} \cdot \beta_{1j} + \beta_0 + \gamma_{jm} + \epsilon_{ijm} \tag{4}$$

where $\gamma_{jm}$ represents the gene-effect on the amount of the TCR feature of interest for SNP $j$ and gene-allele-group $m$, and $\epsilon_{ijm}$ is the random error for subject $i$, SNP $j$, and gene-allele-group $m$ such that $\epsilon_{ijm} \sim \mathcal{N}(0, \sigma^2)$. The variables $x_{ij}$, $\beta_{1j}$, and $\beta_0$ are defined as in the 'simple model' description (*Equation 1*) in the previous section. However, since each subject had a different number of TCRs measured and varying *TCRB* gene usage, we calculated the proportion of TCRs from each candidate *TCRB* gene allele group, $m$, to define a weight, $W_{im}$, for each observation:

$$W_{im} = \frac{N_{im}}{\sum_{m=1}^{M_t} N_{im}}.$$

With this, we solved the following weighted least squares problem for each SNP, TCR feature, and productivity status combination:

$$(\hat{\beta}_0, \hat{\beta}_{1j}, \hat{\gamma}_j) = \text{argmin}_{\beta_0, \beta_{1j}, \gamma_{\cdot j}} \sum_{i=1}^{n} \sum_{m=1}^{M_t} W_{im} \cdot \left( \bar{y}_{im} - (\beta_0 + \gamma_{jm} + \beta_{1j} x_{ij}) \right)^2 \tag{5}$$

using the `lm` function in R.

With each estimate of the $j$-th SNP effect on the amount of the TCR feature of interest, $\hat{\beta}_{1j}$, generated using the models described above, we quantified the association strength between each SNP and the amount of the TCR feature by testing whether $\hat{\beta}_{1j} = 0$. To do this, we applied a t-test (described in the previous section) using the test statistic (*Equation 3*) to obtain each p-value. However,

because our condensed dataset contains a total of $M_t$ observations from each subject $i$, these p-values may be inflated due to intra-subject observations being potentially correlated. Thus, to increase the accuracy of the p-value calculation, for each association p-value below a certain threshold (we chose $P < 5 \times 10^{-5}$), we recalculated the p-value using a clustered bootstrap (with subjects as the sampling unit). To do so, for each bootstrap iterate, we resampled subjects from the condensed dataset with replacement. Using this re-sampled data, we fit the model in *Equation 5* to estimate each coefficient. We repeated this bootstrap process 100 times and used the resulting 100 coefficient estimates to estimate a standard error for each model coefficient. With this re-calculated standard error of the estimate of the $j$-th SNP effect on the amount of the TCR feature of interest, $\text{se}(\hat{\beta}_{1j})$, we wanted to test whether $\hat{\beta}_{1j} = 0$ by recalculating the test-statistic, *Equation 3*, and applying a t-test to obtain each 'corrected' p-value. As noted in the multiple testing correction methods section, when accounting for multiple testing via Bonferroni correction, we used the entire number of TCR features and SNPs considered (not just those that were sufficiently promising to warrant use of the bootstrap to get a more accurate p-value): This ensures that our correction will not be anti-conservative.

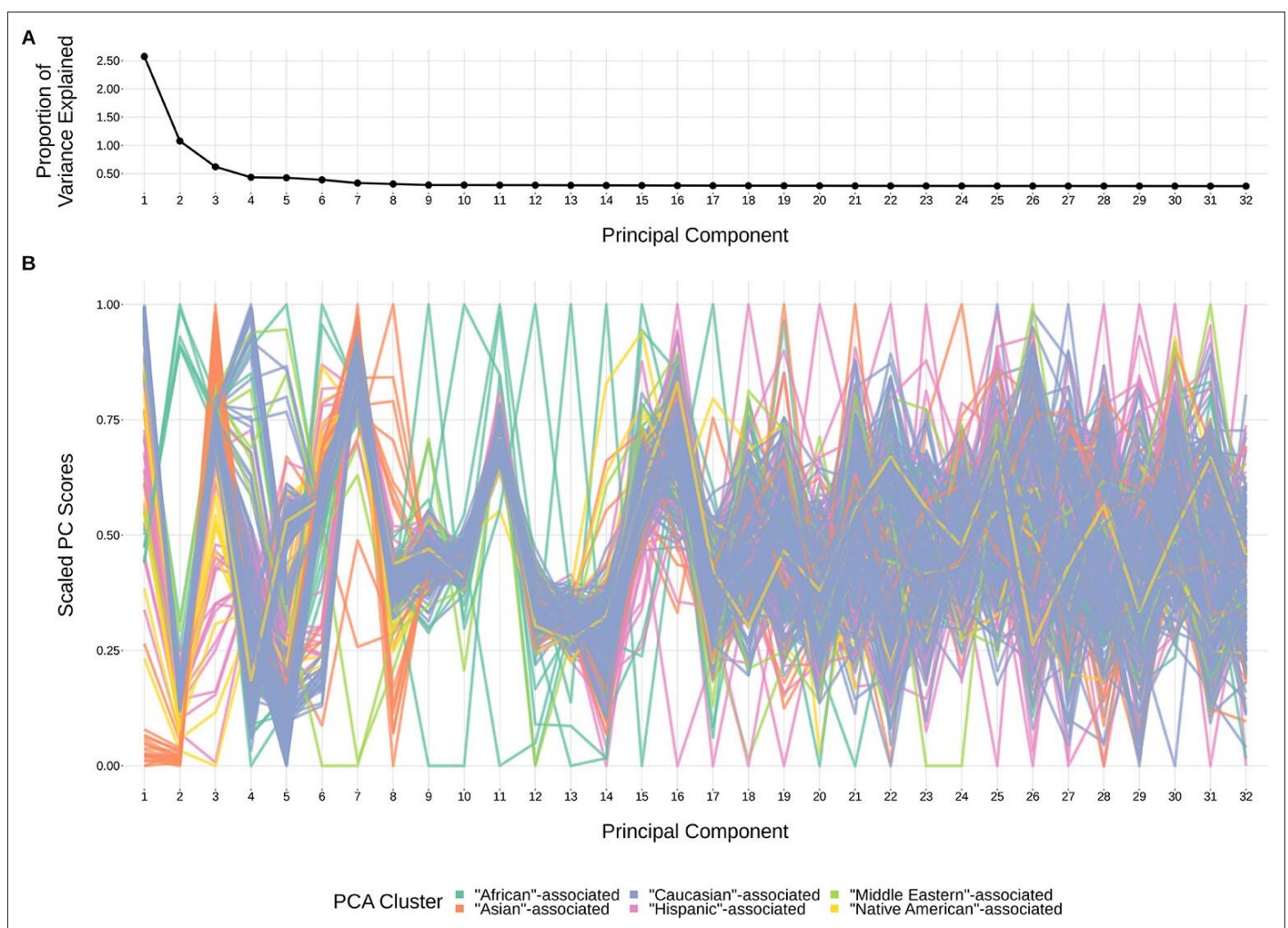

**Figure 9.** The top principal components calculated from genotype data reflect ancestry structure among samples. (**A**) The majority of the ancestry-informative principal component analysis variance is explained by the first eight principal components. (**B**) The first eight principal components show distinct separation by PCA cluster. Each colored line represents one of the 398 samples. The first 32 principal components are shown on the X-axis and their scaled component values for each subject on the Y-axis.

The online version of this article includes the following source data for figure 9:

**Source data 1.** Percent variance explained by each principal component.

**Source data 2.** Scaled principal component values by subject.

## Correcting for population-substructure-related effects

Structure within our SNP genotype data (such as population-substructure-related biases due to ancestry), if present, may produce false positive associations when quantifying the association strength between each SNP and our phenotype of interest. To account for this, we implemented principal component analysis as commonly applied to genome-wide genotype data for population substructure inference. Specifically, we used the PC-AiR algorithm (*Conomos et al., 2015*) which identifies principal components that capture ancestry while accounting for relatedness in the samples. As such, the top principal components calculated from the genotype data reflect population substructure among the samples. When plotting the proportion of variance explained by each PC, we find that the majority of variability appears to be explained by the top eight PCs (*Figure 9*). This conclusion is supported when plotting each PC score by ancestral group (*Figure 9*). With this, we incorporated the top eight principal components as covariates into our GWAS models described above.

As such, to quantify the association strength between each SNP and TCR feature without conditioning on gene usage as in *Equation 1*, while incorporating principal component terms to correct for population-substructure-related bias due to ancestry, we fit the model:

$$\bar{y}_i = x_{ij} \cdot \beta_{1j} + \beta_0 + \sum_{p=1}^{8} \beta_{2jp} \cdot P_{ip} + \epsilon_{ij} \tag{6}$$

where $\bar{y}$, $x_{ij}$, $\beta_{1j}$, $\beta_0$ and $\epsilon_{ij}$ are defined as in *Equation 1*, $\beta_{2jp}$ is the population-substructure-related bias correction term for SNP $j$ and the $p$-th principal component, and $P_{ip}$ is the $p$-th principal component for subject $i$ as calculated above. To estimate each regression coefficient, we solved the following least squares problem for each SNP, TCR feature, and productivity status combination:

$$(\hat{\beta}_0, \hat{\beta}_{1j}, \vec{\hat{\beta}}_{2j}) = \text{argmin}_{\beta_0, \beta_{1j}, \vec{\beta}_{2j}} \sum_{i=1}^{n} \left( \bar{y}_i - (x_{ij} \cdot \beta_{1j} + \beta_0 + \sum_{p=1}^{8} \beta_{2jp} \cdot P_{ip}) \right)^2.$$

Furthermore, to quantify the association strength between each SNP and TCR feature, conditional on gene usage as in *Equation 4*, while incorporating principal component terms to correct for population-substructure-related bias due to ancestry, we fit the model:

$$\bar{y}_{im} = x_{ij} \cdot \beta_{1j} + \beta_0 + \gamma_{jm} + \sum_{p=1}^{8} \beta_{2jp} \cdot P_{ip} + \epsilon_{ijm} \tag{7}$$

where $\bar{y}_{im}$, $x_{ij}$, $\beta_{1j}$, $\beta_0$, $\gamma_{jm}$ and $\epsilon_{ij}$ are defined as in *Equation 4* and $\hat{\beta}_{1j}$ and $P_{ip}$ are defined as in *Equation 6*. Again, to estimate each regression coefficient, we solved the following weighted least squares problem for each SNP, TCR feature, and productivity status combination:

$$(\hat{\beta}_0, \hat{\beta}_{1j}, \hat{\gamma}_j, \vec{\hat{\beta}}_{2j}) = \text{argmin}_{\beta_0, \beta_{1j}, \gamma_j, \vec{\beta}_{2j}} \sum_{i=1}^{n} \sum_{m=1}^{M_t} W_{im} \cdot \left( \bar{y}_{im} - (\beta_0 + \gamma_{jm} + \beta_{1j} x_{ij} + \sum_{p=1}^{8} \beta_{2jp} \cdot P_{ip}) \right)^2.$$

With these estimates for the population-substructure-corrected $j$-th SNP effect on the amount of the TCR feature of interest, $\hat{\beta}_{1j}$, we calculated a P-value using the methods described in the methods section for each model type.

## Correcting for *TRBD2* allele genotype, SNP genotype linkage when quantifying SNP, TCR feature associations within the *TCRB* locus

Within the *TCRB* locus, we noted that SNP genotypes were associated with *TRBD2* allele genotype (*Figure 3—figure supplement 1*). Associations between gene-alleles and *TCRB* locus SNP genotypes, if present, may produce false positive associations when implementing the 'gene-conditioned model' to infer associations between SNPs and TCR repertoire features, conditional on gene usage. To explore this phenomenon further, we zoomed in to the *TCRB* locus and incorporated a *TRBD2* allele genotype correction procedure into our model formulation. As such, to quantify the association strength between each *TCRB* locus SNP and TCR feature, conditional on gene usage and correcting for population-substructure-related effects as in *Equation 7*, while incorporating *TRBD2* allele genotype correction terms, we fit the model:

$$\bar{y}_{im} = z_i \cdot \alpha_j + x_{ij} \cdot \beta_{1j} + \beta_0 + \gamma_{jm} + \sum_{p=1}^{8} \beta_{2jp} \cdot P_{ip} + \epsilon_{ijm}$$

where $z_i$ represents the qualitative *TRBD2* allele genotype status for subject $i$ such that $z_i \in$ {"TRBD2*01 homozygous", "heterozygous", "TRBD2*02 homozygous"}, $\alpha_j$ is the *TRBD2* allele genotype effect for SNP $j$, and the remaining variables are defined as in *Equation 7*. With this model formulation, we can estimate each regression coefficient by solving the following weighted least squares problem for each *TCRB* SNP, TCR feature, and productivity status combination:

$$(\hat{\alpha}_j, \hat{\beta}_0, \hat{\beta}_{1j}, \hat{\gamma}_j, \vec{\hat{\beta}}_{2j}) = \mathrm{argmin}_{\alpha_j, \beta_0, \beta_{1j}, \gamma_j, \vec{\beta}_{2j}} \sum_{i=1}^{n} \sum_{m=1}^{M_t} W_{im} \cdot \left(\bar{y}_{im} - (\alpha_j z_i + \beta_0 + \gamma_{jm} + \beta_{1j} x_{ij} + \sum_{p=1}^{8} \beta_{2jp} \cdot P_{ip})\right)^2.$$

With these estimates for the *TRBD2* allele genotype and population-substructure-corrected $j$-th SNP effect on the amount of the TCR feature of interest, $\hat{\beta}_{1j}$, we calculated a p-value using the methods described in the Materials and methods section for the 'gene-conditioned model'.

## Multiple testing correction for associations

For each TCR feature (i.e. extent of trimming, number of N-insertions, etc.), we considered the significance of associations using a Bonferroni-corrected threshold. To establish each threshold, we corrected for each TCR feature subtype (i.e. V-gene trimming, J-gene trimming, etc. for the TCR trimming feature), the two TCR productivity types (productive and non-productive), and the total number of SNPs tested. When considering associations in the whole-genome context, we corrected for the approximately 6.5 million SNPs (remaining after filtering). When considering associations in a gene-level context, we corrected for the number of SNPs within 200 kb of the gene of interest. For the validation analysis, we considered associations in a SNP-level context and did not correct for multiple SNPs. However, for the validation analysis, we considered the significance of associations within both TCRα and TCRβ chains and, thus, corrected the significance threshold accordingly.

## Genomic inflation factor calculations

We defined the genomic inflation factor $\lambda$ to be the ratio of the median of the empirically observed squared test statistic to the expected median (*Devlin and Roeder, 1999*; *Freedman et al., 2004*; *Price et al., 2010*). For each GWAS analysis implemented using the 'simple model', we used the test statistic $T_j$ given by *Equation 3* for each SNP $j = \{1 \ldots J\}$ tested genome-wide. For each GWAS analysis implemented using the 'gene-conditioned model', it was not computational feasible to calculate a test statistic $T_j$ for all SNPs tested genome-wide using the bootstrapping protocol described in the 'gene-conditioned model' Materials and methods section. Thus, instead, we randomly sampled 10,000 SNPs and calculated the test statistic $T_j$ for each SNP in the random subset. Let $S = \{T_1^2, \ldots, T_J^2\}$ be the set of all squared test statistics. As such,

$$\lambda = \frac{\mathrm{median}(S)}{0.456}$$

where 0.456 is the median of a chi-squared distribution with one degree of freedom. If the GWAS analysis results follow the chi-squared distribution, the expected value of $\lambda$ is 1. Thus, when $\lambda < 1.03$, we concluded that there was no evidence of systemic population-substructure-related bias in the analysis (*Price et al., 2010*; *Conomos et al., 2016*).

## Conditional analysis to test for multiple independent association signals

Within the *DNTT* and *DCLRE1C* loci, we performed a stepwise series of nested regression analyses to test for independent SNP associations within each locus for N-insertion and nucleotide trimming, respectively. We used the same models and covariates as the primary analyses ('simple model' for associations between N-insertion and *DNTT* variation and the 'gene-conditioned model' for associations between nucleotide trimming and *DCLRE1C* variation) and included the most significant SNP within each locus as an additional covariate. We inferred the association between each SNP within each locus and the TCR feature of interest using this new conditional model and considered significant associations at a gene-level Bonferroni-corrected significance threshold for each locus. From here, we repeated this analysis (if necessary), identifying and adding additional SNPs one-by-one as a covariate to each successive model. Once the p-value of top SNP within the locus was no longer significant, we concluded the analysis. SNPs which were added as as additional covariates in the final conditional model were considered to be independent signals.

## Ancestry-informative PCA cluster classification

In order to correct for population-substructure-related biases due to ancestry in our GWAS analyses, we used ancestry-informative principal component analysis. The original genotyping dataset (*Martin et al., 2020*) contained self-reported ancestry. However, a number of subjects did not self-report ancestry in the original data collection. Further, for some subjects, their self-reported ancestry was discordant with clusters observed in a principal component analysis. Therefore, for analysis purposes, we used the minimum covariance determinant method (*Rousseeuw and Driessen, 1999*; *Conomos et al., 2016*) with the original self-identified labels to group the subjects into six ancestry-informative PCA clusters: 'African'-associated (8), 'Asian'-associated (23), 'Caucasian'-associated (322), 'Hispanic'-associated (30), 'Middle Eastern'-associated (5), and 'Native American'-associated (10).

## Quantifying associations between *TRBD2* allele genotype and SNP genotype within the *TCRB* locus

For each significantly associated SNP within the *TCRB* locus as shown in *Figure 3*, we compared SNP genotype to *TRBD2* allele genotype across all subjects. We used Pearson correlation to measure the correlation between the two genotypes.

## Quantifying TCR repertoire feature and SNP minor allele frequency variations by ancestry-informative PCA cluster

To quantify PCA cluster variation of TCR repertoire features (such as total N-insertions [V-D N-insertion and D-J N-insertion]), we first calculated an average of each TCR repertoire feature by subject and productivity status. We also calculated a population mean of each TCR repertoire feature by productivity status. Each subject was classified into one of six PCA clusters. Thus, we compared the mean of the TCR repertoire features within each PCA cluster to the population mean using a one-sample t-test to compute each P-value. We used Bonferroni multiple testing correction to adjust each p-value.

We also calculated SNP minor allele frequencies for the whole population and for each PCA cluster individually such that

$$\mathrm{MAF}_{jr} = \frac{\sum_{i=1}^{I_r} x_{ij}}{2 * I_r}. \tag{8}$$

Here, $\mathrm{MAF}_{jr}$ is the minor allele frequency for SNP marker $j$ and PCA cluster $r$, $I_r$ is the number of individuals in the PCA cluster $r$, and $x_{ij}$ is the number of alleles in the genotype of SNP marker $j$ for subject $i \in \{1, \ldots, I_r\}$. For each SNP $j$, the minor allele was defined as the allele with the lowest frequency in the total population. To quantify minor allele frequency differences by PCA cluster for select SNPs within various loci of interest (i.e. *DNTT* gene), we compared the minor allele frequencies calculated within PCA-clusters to the minor allele frequencies calculated for the entire population using a one-sample t-test to compute each P-value. Again, we used Bonferroni multiple testing correction to adjust each p-value.

For both of these analyses, we used the `t_test` function from the `rstatix` package in R.

## Implementation and Code

R code implementing the genome-wide association inferences described here is available at https://github.com/phbradley/tcr-gwas, (copy archived at swh:1:rev:fd4f43562a63d45721d61f54d99d4bc493cb4066; *Russell, 2022*). The following tools were especially helpful:

- data.table (*Dowle and Srinivasan, 2021*)
- tidyverse (*Wickham et al., 2019*)
- doParallel (*Corporation and Weston, 2020*)
- SNPRelate (*Zheng et al., 2012*)
- GWASTools (*Gogarten et al., 2012*)
- GENESIS (*Gogarten et al., 2019*)
- cowplot (*Wilke, 2020*)

## Acknowledgements

The authors thank Christopher Carlson, William DeWitt, and Michael Lieber for helpful discussions regarding this paper. The authors would also like to thank Fred Hutch scientific computing (National Institutes of Health, ORIP S10OD028685). Dr. Matsen is an Investigator of the Howard Hughes Medical Institute.

## Additional information

### Competing interests

Aubree Gordon: serves on a scientific advisory board for Janssen. Paul G Thomas: consults for Johnson and Johnson, Immunoscape, Cytoagents, and PACT Pharma. He has received travel reimbursement from 10X Genomics and Illumina. He is an inventor on two pending US patent applications related to T cell receptor biology (US: 15/780,938 titled "Cloning and Expression System for T-Cell Receptors' and US: 17/616,279 titled "Kit and Method for Analyzing Singlet Cells'). The other authors declare that no competing interests exist.

### Funding

| Funder | Grant reference number | Author |
|---|---|---|
| National Institutes of Health | R01 AI146028 | Magdalena L Russell<br>Noah Simon<br>Frederick A Matsen IV<br>Philip Bradley |
| National Institutes of Health | R01 AI136514 | Aisha Souquette<br>Stefan A Schattgen<br>E Kaitlynn Allen<br>Paul G Thomas<br>Philip Bradley |
| National Institutes of Health | R01 AI120997 | Guillermina Kuan<br>Angel Balmaseda<br>Aubree Gordon |
| National Institutes of Health | R01 AI107625 | Aisha Souquette<br>Stefan A Schattgen<br>E Kaitlynn Allen<br>Paul G Thomas |
| National Institute of Allergy and Infectious Diseases | HHSN272201 400006C | Aisha Souquette<br>Stefan A Schattgen<br>E Kaitlynn Allen<br>Guillermina Kuan<br>Angel Balmaseda<br>Aubree Gordon<br>Paul G Thomas |
| National Institute of Allergy and Infectious Diseases | 75N93021C00016 | Aisha Souquette<br>Stefan A Schattgen<br>E Kaitlynn Allen<br>Guillermina Kuan<br>Angel Balmaseda<br>Aubree Gordon<br>Paul G Thomas |
| National Institute of Allergy and Infectious Diseases | AI33484 | David M Levine |
| National Institute of Allergy and Infectious Diseases | AI149213 | David M Levine |
| National Cancer Institute | CA015704 | David M Levine |
| National Heart, Lung, and Blood Institute | HL087690 | David M Levine |
| National Heart, Lung, and Blood Institute | HL088201 | David M Levine |

| Funder | Grant reference number | Author |
|---|---|---|
| National Heart, Lung, and Blood Institute | HL094260 | David M Levine |
| National Heart, Lung, and Blood Institute | HL105914 | David M Levine |
| National Heart, Lung, and Blood Institute | K23HL69860 | David M Levine |
| The Simons Foundation and Howard Hughes Medical Institute | 55108544 | Frederick A Matsen IV |
| Howard Hughes Medical Institute | Investigator | Frederick A Matsen IV |

The funders had no role in study design, data collection and interpretation, or the decision to submit the work for publication.

### Author contributions

Magdalena L Russell, Formal analysis, Methodology, Software, Visualization, Writing – original draft, Writing – review and editing; Aisha Souquette, Data curation, Investigation, Writing – review and editing; David M Levine, Methodology, Writing – review and editing; Stefan A Schattgen, Data curation, Investigation; E Kaitlynn Allen, Data curation, Investigation, Resources; Guillermina Kuan, Noah Simon, Data curation, Investigation, Methodology, Resources, Writing – review and editing; Angel Balmaseda, Data curation, Funding acquisition, Investigation, Resources, Supervision; Aubree Gordon, Data curation, Funding acquisition, Resources, Supervision, Writing – review and editing; Paul G Thomas, Frederick A Matsen, Philip Bradley, Conceptualization, Data curation, Formal analysis, Funding acquisition, Methodology, Resources, Software, Supervision, Writing – original draft, Writing – review and editing

### Author ORCIDs

Magdalena L Russell ![ORCID] http://orcid.org/0000-0002-1068-1968
Frederick A Matsen IV, ![ORCID] http://orcid.org/0000-0003-0607-6025
Philip Bradley ![ORCID] http://orcid.org/0000-0002-0224-6464

### Ethics

For the validation cohort, participants provided written informed consent and parental permission was obtained from parents or legal guardians of children, in addition to verbal assent from children aged six years and older. This study was approved by the Institutional Review Boards at the University of Michigan (HUM 00091392) and the Centro Nacional de Diagnóstico y Referencia (Ministry of Health, Nicaragua; CIRE 06/07/10-025).

### Decision letter and Author response

Decision letter https://doi.org/10.7554/eLife.73475.sa1
Author response https://doi.org/10.7554/eLife.73475.sa2

## Additional files

### Supplementary files

• Transparent reporting form

### Data availability

Validation cohort TCRA- and TCRB-immunosequencing data have been deposited into The BioProject database under accession code PRJNA762269. Validation cohort SNP data have been deposited into the Zenodo database (DOI:10.5281/zenodo.5719516). Discovery cohort SNP array data are previously published and are available in The database of Genotypes and Phenotypes under accession code phs001918. Discovery cohort TCRB-immunosequencing data are also previously published and are available in the ImmuneAccess database (DOI:10.21417/B7001Z). All data generated or analysed during this study are included in the manuscript and supporting files; Source Data files have

been provided for Figures 1, 2, 3, 4, 5, 6, 7, 8, and 9. All data processed during this study have been deposited in the Zenodo database (discovery cohort data available at DOI:10.5281/zenodo.5719520 and validation cohort data available at DOI:10.5281/zenodo.5719516). Code implemented in this study has been made available on GitHub:https://github.com/phbradley/tcr-gwas, (copy archived at swh:1:rev:fd4f43562a63d45721d61f54d99d4bc493cb4066).

The following datasets were generated:

| Author(s) | Year | Dataset title | Dataset URL | Database and Identifier |
|---|---|---|---|---|
| Souquette A, Schattgen SA, Kuan G, Balmaseda A, Gordon A, Thomas PG | 2021 | The Nicaraguan Influenza Cohort Study | https://www.ncbi.nlm.nih.gov/bioproject/PRJNA762269 | NCBI BioProject, PRJNA762269 |
| Souquette A, Schattgen SA, Allen EK, Kuan G, Balmaseda A, Gordon A, Thomas PG | 2021 | Combining genotypes and T cell receptor distributions to infer genetic loci determining V(D)J recombination probabilities: validation cohort meta data and parsed TCR repertoire data | https://doi.org/10.5281/zenodo.5719516 | Zenodo, 10.5281/zenodo.5719516 |
| Levine DM, Bradley P | 2021 | Combining genotypes and T cell receptor distributions to infer genetic loci determining V(D)J recombination probabilities: discovery cohort meta data and parsed TCR repertoire data | https://doi.org/10.5281/zenodo.5719520 | Zenodo, 10.5281/zenodo.5719520 |

The following previously published datasets were used:

| Author(s) | Year | Dataset title | Dataset URL | Database and Identifier |
|---|---|---|---|---|
| Emerson RO, DeWitt WS, Vignali M, Gravley J, Osborne EJ, Desmarais C, Klinger M, Carlson CS, Hansen JA, Rieder M, Robins HS | 2017 | Immunosequencing identifies signatures of cytomegalovirus exposure history and HLA-mediated effects on the T-cell repertoire | https://doi.org/10.21417/B7001Z | ImmuneACCESS, 10.21417/B7001Z |
| Martin PJ, Levine DM, Storer BE, Nelson SC, Dong X, Hansen JA | 2020 | STAMPEED: Whole Genome Association Analysis of Hematopoietic Cell Transplant (HCT) Outcomes | https://www.ncbi.nlm.nih.gov/projects/gap/cgi-bin/study.cgi?study_id=phs001918.v1.p1 | NCBI dbGaP, phs001918.v1.p1 |

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
