## [Editor Report]

This study demonstrates that genetic differences in areas of the genome outside the regions that encode the TCR genes can affect the molecular properties of the TCRs that are made by somatic recombination. This paper will be of interest to a broad swathe of immunologists who study such variable lymphocyte receptors. It combines several large datasets in an extremely statistically rigorous analysis, producing results consistent with but substantially expanding upon the prior knowledge of the field.

---

## [Decision Letter]

**Decision letter after peer review:**

Thank you for submitting your article "Combining genotypes and T cell receptor distributions to infer genetic loci determining V(D)J recombination probabilities" for consideration by *eLife*. Your article has been reviewed by 3 peer reviewers, including Benny Chain as Reviewing Editor and Reviewer #1, and the evaluation has been overseen by Aleksandra Walczak as the Senior Editor. The following individual involved in review of your submission has agreed to reveal their identity: James M. Heather (Reviewer #2).

Essential revisions:

Please respond to the specific issues raised by reviewers 2 and 3.

*Reviewer #1 (Recommendations for the authors):*

The genetic association data seems mostly convincing.

In addition to the Manhattan plots, it would be very informative to see the actual effects on variable gene freqeuncy, their magnitude and the extent to which different genes are regulated together – e.g. do some SNPs regulate several different Vs together? Similarly, more detail on the magnitude and details of the effects seen on basde pair deletion and addition would perhaps contribute to a better understnading of the importance and impact of the genetic regulation.

Weaknesses of the study: 1. Limited additional understanding of the process of regulation of T cell receptors; limited validation of results in an independent data sets.

Overall, the findings are sound, but of incrememental importance in understanding TCR repertoire generation. A more specialised human genetics journal may be a more appropriate place to publish these results, and the methodology for conditioning one outcome on another outcome, rather than the more general readership of *eLife*.

*Reviewer #2 (Recommendations for the authors):*

This manuscript is already extremely well put together, making it a genuine pleasure to read, so I have only a few recommendations.

(1a) I think it may be useful to have the ZNF peaks on Fig 1 labelled in some way. They definitely stand out when you the paper, yet are not addressed in the narrative of the paper until a figure or two later.

(1b) On a related note, unless I missed it, it's probably worth explicitly mentioning that the TRVB24-1 association with ZNF443 was also observed in the Sharon et al paper too.

(2) Fig 3 seems to show some signal for a weaker but significant SNP association with trimming on chr23. This is another observation that leaps out to the reader, but one for which I couldn't find a mention of in the text at all.

(3) I appreciate that I may be unwittingly stirring up a well-trodden semantics issue from a field not my own, but would not many of these associations be considered eQTL? At the very least those involving productive gene expression as shown in Fig 1. If so I think it might be helpful to include the term in the text somewhere, even if only buried in the methods, purely to make this paper more likely to be returned in relevant literature searches.

(4) I'm not sure how practical a request this is, but given the discussion of SNP coverage in the public review section: is there any way for the authors to measure (or even speculate) as to what portion of known TCR loci polymorphisms are covered by the SNP arrays used in these datasets? It feels like a number that might be helpful for contextualising the results of this study.

(5) Lines 156/157 report that various SNPs associated with the expression "of the V-genes TRBV24-1 and TRBV24/OR9-2". A very minor pedantic point, but this sentence does make it sound like the expression of both genes (or expression of recombinations using both genes) is going up. Seeing as TRBV24/OR9-2 resides on another chromosome, it's almost certain that in actuality it's just that TRBV24-1 TCRs are increasing and the orphon gene recombination levels are remaining steady around zero. Incidentally this is a good illustration of a case where the short Adaptive reads can't discriminate between two genes, even when the immunological importance differs widely.

(6a) While I didn't have time to actually run the code, I did note that the first step in the README is a little vague ("Download data into the directory ..."). A little more instruction would be very useful here, as in my experience half the battle in getting other people's analyses running is matching input data formats. Obviously the manuscript lists the accessions of the data themselves, but explicit discussion of the pre-processing steps would be extremely helpful for anyone wanting to re-run or adapt these analyses. (I see that some information is planned to be included on Zenodo prior to publication, so if this is to be included there please disregard this comment.)

(6b) On a related note, I noticed that a few details relevant for repeatability have been omitted from the methods. In particular the versions and non-default parameters of software tools used (e.g. for the TCRdist and MiGEC VDJ analyses), and the date of accession of databases (e.g. IMGT/GENE-DB) should be included. Given the nature of this manuscript and the frequency of changes to GENE-DB I would even recommend actually uploading the specific version(s) of the database that were used.

(7) The lines of Fig 9B are very narrow, which makes it very hard to tell the difference between some of the groups (especially in the legend). Perhaps the lines could be made wider, or some lines made dotted or dashed or something, so as to make the groups easier to distinguish.

(8) The observation that the different ancestry-associated groups differ in some of their recombination parameters is very interesting: I can't recall seeing similar data before, beyond some general V gene expression level differences (typically thought to be a consequence of differing HLA allele distributions). However, having never performed such an ancestry-informative PCA before I have what may be naive concerns about it. For instance, of the ~400 people in the cohort, it seems that all are assigned to one of the groups, while I would presume that out of those 400 people there are likely some multiracial individuals or those who just fall out of the typical expected SNP distributions (which Fig 9 would suggest is the case). Some more discussion of this issue may be instructive for those like myself who have never run such an analysis. For example, are those individuals who are further from the center of their respective clusters - or perhaps those whose clustered ancestry group is different to their self-reported one - enriched in the outliers in Figs 7 and 8? Similarly, I'm curious as to how many individuals in each group end up assigned to another, and to which. These considerations seem especially important given the different sizes of the groups (with the 'Caucasian' associated group having hundreds of individuals, and the other groups mostly having fewer than ten), and the fact that the target clusters are themselves informed by the original input reported groupings. While obviously beyond the scope of this manuscript, it's interesting in itself to note that such an observation hasn't been made before. Presumably most TCRseq studies are not large or diverse enough to have detected such differences (even had people been looking)?

*Reviewer #3 (Recommendations for the authors):*

1. Regarding the datasets, full information regarding the donors from which the data have been used need to be summarized in a Table and couple of figures to identify possible (or no if this is the case) bias in terms of sex, age, ethnicity, influenza exposure in the days before the sample collection, CMV serotype, etc… Indeed, all these factors can influence the repertoire composition, and maybe some correction/normalization should be applied.

2. On the source data table containing 9957 associations, 43 TRBV out the 66 mentioned showed significant associations with 1 to 508 SNPs. For several TRBV genes, the association with SNPs was significantly different according to the productive vs nonproductive origin of the TRBV in the dataset. Same for the 10 TRBJ (out of the 14 tested) showing significant associations with SNPs.

First, the "significant association" column in Table 1 should reflect those results by indicating the number of V genes and J genes found to be associated with at least one SNP above the significance threshold.

Second, in the data source table we can see some V gene usage (and J gene usage as well) from nonproductive and productive rearrangements are associated with various number of SNPs. What is the overlap of the SNPs associated with each V according to its nonproductive/productive origin? In addition, a general comment on the use of nonproductive rearrangement data. Such "part" of the TCR repertoire is believed to reflect TCR generation independently of TCR selection. However, when analysed from blood TCR repertoires (like in this study), it is still unclear how much the nonproductive "repertoire" is biased by the fact that it is directly dependent on the productive, and therefore centrally/peripherally selected. Therefore, the variations associated at the V, J usage (as well as the trimming) may be biased by the immune history of every individual. Author should control on this possible bias or disregard the differences between productive vs. nonproductive sequences. The only dataset that could help address this question would be thymic cells at the different stage of differentiation.

Third, in the text it is indicated that "variation in the TCRB locus is most significantly associated with the expression of the gene TRBV28 for both the productive and nonproductive". Unless I missed something, this is also (maybe more) true for TRBV12-3, TRBV12-4 (known to be highly expressed in general) as well as TRBV24-1, accounting for respectively 353, 358 and 319 SNP associations compared with 424. Are those differences significant? Are you referring to TRBV28 for a particular reason (major overlap between SNPs associations between nonproductive and productive for instance or something else)? This should be clarified and detailed.

3. On the association between HLADRB1 and TRBV10-3, the authors refers to two other studies that found such association however they omit to discuss the fact that on those studies they found in fact other TRBV gene usage to be much more associated with the HLADRB1, notably TRBV20-1 (Gao et al., 2019) which is not found from the Emerson dataset. Are the differences associated with the ethnicity (as Gao paper is mainly done on Asian population)? Maybe to provide some functional relationship of the associations, it could be of interest to analyze data from patients with AIDs, such as RA, SLE, Sjogren syndrome known to be associated with HLADRB1. Data from the Rossetti paper Rosseti et al., (Annals of Rheum Dis, 2017; TCRb data from Adaptive available online) as well as from the Liu et al. (Annals of Rheum Dis, 2019). For instance it could be of interest to determine whether in RA or SLE patients, a differential usage of TRBV10-3, TRBV20 compared to controls has been shown. Eventually, if DNA is still available, ensure the HLADRB1 genotype to correlate with the observations.

4. Regarding the SNPs association with the gene trimming and N-insertion numbers, interestingly the genes showing SNPs association with this TCR repertoire feature are definitely biologically linked. However, although the author distinguish the impact of the gene and on the trimming versus N-insertion, since the resulting repertoire analyzed is a post-selection repertoire, the observation are still bias by the selection effect. Moreover, it is also well known that shorted TCRs are more frequent in general, than long ones. In other word, authors should control for these bias and provide more evidence on the actual SNPs identified between the discovery and the validation cohorts.

---

## [Author Response]

Reviewer #1 (Recommendations for the authors):The genetic association data seems mostly convincing.In addition to the Manhattan plots, it would be very informative to see the actual effects on variable gene freqeuncy, their magnitude and the extent to which different genes are regulated together – e.g. do some SNPs regulate several different Vs together? Similarly, more detail on the magnitude and details of the effects seen on basde pair deletion and addition would perhaps contribute to a better understnading of the importance and impact of the genetic regulation.

We have added details about the magnitude of the effects for the significant associations observed for gene-usage within the MHC and TCRb loci and for nucleotide trimming and N-insertion within the DCLRE1C and DNTT loci, respectively.

For associations observed for gene-usage within the TCRb locus, we have added lines 139-142:

"For the significantly associated TCRB locus SNPs, the median association effect magnitude was largest for the expression of TRBD1 (median effect size = -0.038) followed by the expression of TRBD2 (median effect size = 0.035) and the expression of TRBV28 (median effect size = 0.019) all in productive TCRs."

For associations observed for gene-usage within the MHC locus, we have added lines 153-156:

"For the significantly associated MHC locus SNPs, the median association effect magnitude was largest for the expression of TRBV4-1 (median effect size = -0.004) followed by the expression of TRBV10-3 (median effect size = 0.0033)."

For associations observed for nucleotide trimming within the DCLRE1C locus, we have added lines 199-201:

"For these significant DCLRE1C locus SNP associations, the magnitudes of the effects were greater for non-productive TCRs compared to productive TCRs for both V-gene trimming and J-gene trimming."

For associations observed for N-insertion within the DNTT locus, we have added lines 271-273:

"For these significant DNTT locus SNP associations, the magnitudes of the effects were greater for non-productive TCRs compared to productive TCRs for both V-D-gene junction N-insertion and D-J-gene junction N-insertion."

We have also added corresponding supplementary figures (Figure 1—figure supplement 1 and 2, Figure 4—figure supplement 1, Figure 6—figure supplement 1).

Weaknesses of the study: 1. Limited additional understanding of the process of regulation of T cell receptors; limited validation of results in an independent data sets.Overall, the findings are sound, but of incrememental importance in understanding TCR repertoire generation. A more specialised human genetics journal may be a more appropriate place to publish these results, and the methodology for conditioning one outcome on another outcome, rather than the more general readership of eLife.

Thank you for you thoughts, however, we like *eLife* because of its general readership and focus on open science. We provide the first examples of genetic variants which are associated with modifying the extent of nucleotide trimming and N-insertion at V(D)J-gene junctions, and because V(D)J-gene junctional diversity is a major source of the overall diversity within the TCR repertoire, we believe our results are important for continuing to understand the T cell receptor generation process. Association between N-insertions and genetic variants near TdT (which likely influence its expression level) explain previous findings of population-level variation in N-insertions (Rubelt et. al., Nature Communications 2015 and Sethna et. al., PLoS Computational Biology 2020) and fit with known effects of TdT expression level on N-insertions during development. Our finding that genetic variation in and around the DCLRE1C locus (which codes for Artemis) influences the extent of nucleotide trimming, is both exceptionally strong statistically (highly significant P values in both the discovery and validation cohorts), and intriguing from a mechanistic perspective.

According to an expert in V(D)J recombination, our current understanding of Artemis regulation would not predict that simple variation in its expression level would have a significant effect on nucleotide trimming (M. Lieber, pers. comm). Thus, we believe that the very strong Artemis associations we find represent more than a simple and incremental extension of our current understanding of V(D)J recombination; they may have important mechanistic implications that lead to new insights.

It's also worth pointing out that these cohorts (N=398 and N=94) are quite small from the GWAS perspective, which attests to the strength of the associations we discovered. As such, we believe our results will be interesting for a general immunology and genetics audience, which form part of the *eLife* readership.

Reviewer #2 (Recommendations for the authors):This manuscript is already extremely well put together, making it a genuine pleasure to read, so I have only a few recommendations.(1a) I think it may be useful to have the ZNF peaks on Fig 1 labelled in some way. They definitely stand out when you the paper, yet are not addressed in the narrative of the paper until a figure or two later.

We have added a ZNF label to Figure 1, thanks.

(1b) On a related note, unless I missed it, it's probably worth explicitly mentioning that the TRVB24-1 association with ZNF443 was also observed in the Sharon et al paper too.

We have added the following sentence (lines 169-171 in the manuscript) describing that significant association between variation near ZNF443 and TRVB24-1 expression was also observed in the Sharon et al paper, thanks:

"Significant association between variation near the ZNF443 locus and expression of TRBV24-1 in productive TCRs was also noted previously (Sharon et al., 2016)."

(2) Fig 3 seems to show some signal for a weaker but significant SNP association with trimming on chr23. This is another observation that leaps out to the reader, but one for which I couldn't find a mention of in the text at all.

Despite looking closely, we were unable to find any genes proximal to the significant SNPs on chromosome 23. Because of this, we have chosen to not discuss these associations in the text.

(3) I appreciate that I may be unwittingly stirring up a well-trodden semantics issue from a field not my own, but would not many of these associations be considered eQTL? At the very least those involving productive gene expression as shown in Fig 1. If so I think it might be helpful to include the term in the text somewhere, even if only buried in the methods, purely to make this paper more likely to be returned in relevant literature searches.

We have added the following sentence (lines 418-421 in the manuscript) to the discussion to suggest that further experimentation will be required to determine whether the significant SNPs associated with changing the extent of nucleotide trimming and N-insertion identified here could be acting as eQTLs:

"The significant SNPs associated with changing the extent of nucleotide trimming and N-insertion identified here could be expression quantitative trait loci (eQTLs), however, experimental work will be required to determine whether these SNPs modify the expression of DCLRE1C and DNTT, respectively."

(4) I'm not sure how practical a request this is, but given the discussion of SNP coverage in the public review section: is there any way for the authors to measure (or even speculate) as to what portion of known TCR loci polymorphisms are covered by the SNP arrays used in these datasets? It feels like a number that might be helpful for contextualising the results of this study.

We agree that including a measure of the proportion of known TCR loci polymorphisms which are covered by the SNP array used here would be valuable. The SNP array used here (after imputation) includes 7304 SNPs within the TRB region (hg19:chr7:141950000-142550000), a fact that we have added to the discussion. Given the complexity of the TCRbeta locus, we expect that this is likely to represent only a small fraction of the actual variation present across diverse populations. Cross-referencing these SNPs with known allelic variation in TCRbeta genes is made challenging by inaccuracies in the gene annotations for the hg19 build; our efforts to 'liftover' the SNP dataset to the hg38 build using automated tools suffered from high dropout rates. For these reasons, we were not able to arrive at a quantitative estimate of SNP coverage in the TCRbeta locus. Anecdotally, we note that when looking for TRB SNPs whose gene usage associations were much more significant for productive than non-productive SNPs, we identified 6 of the 7 genes (TRBV12-5, TRBV7-3, TRBV11-1, TRBV11-3, TRBV10-1, and TRBV30) flagged in the Dean et al. study as having both productive and non-productive alleles (PMID 26596423; see Fig. 4). This suggests that our SNPs likely cover many of the known TRB alleles, however since each allele can be associated with variation at multiple nucleotides, panel coverage could still be quite low when measured at the per-nucleotide level.

(5) Lines 156/157 report that various SNPs associated with the expression "of the V-genes TRBV24-1 and TRBV24/OR9-2". A very minor pedantic point, but this sentence does make it sound like the expression of both genes (or expression of recombinations using both genes) is going up. Seeing as TRBV24/OR9-2 resides on another chromosome, it's almost certain that in actuality it's just that TRBV24-1 TCRs are increasing and the orphon gene recombination levels are remaining steady around zero. Incidentally this is a good illustration of a case where the short Adaptive reads can't discriminate between two genes, even when the immunological importance differs widely.

This is a great point. For our gene usage associations, we have now restricted our analyses to non-orphan genes. We have included this change within the Methods section (lines 545-547), and changed the corresponding results accordingly.

(6a) While I didn't have time to actually run the code, I did note that the first step in the README is a little vague ("Download data into the directory ..."). A little more instruction would be very useful here, as in my experience half the battle in getting other people's analyses running is matching input data formats. Obviously the manuscript lists the accessions of the data themselves, but explicit discussion of the pre-processing steps would be extremely helpful for anyone wanting to re-run or adapt these analyses. (I see that some information is planned to be included on Zenodo prior to publication, so if this is to be included there please disregard this comment.)

Thanks for this suggestion. We have included all parsed TCR repertoire data files and all required meta data files on Zenodo (https://doi.org/10.5281/zenodo.5719520 for the discovery cohort and https://doi.org/10.5281/zenodo.5719516 for the validation cohort). Details regarding the pre-processing steps for these parsed TCR repertoire data files have been included within the GitHub README. We have also provided further details about accessing and downloading these data, in addition to details about accessing, downloading, and processing the SNP data from The database of Genotypes and Phenotypes (dbGaP), within the GitHub README.

(6b) On a related note, I noticed that a few details relevant for repeatability have been omitted from the methods. In particular the versions and non-default parameters of software tools used (e.g. for the TCRdist and MiGEC VDJ analyses), and the date of accession of databases (e.g. IMGT/GENE-DB) should be included. Given the nature of this manuscript and the frequency of changes to GENE-DB I would even recommend actually uploading the specific version(s) of the database that were used.

This is a great suggestion. We have added details about the versions and non-default parameters for each software tool used for pre-processing the TCR repertoire data within the GitHub README. We have also uploaded the specific TRB and TRA genes and sequences used from IMGT to Zenodo.

(7) The lines of Fig 9B are very narrow, which makes it very hard to tell the difference between some of the groups (especially in the legend). Perhaps the lines could be made wider, or some lines made dotted or dashed or something, so as to make the groups easier to distinguish.

We have made Figure 9B larger and increased the line widths, thanks.

(8) The observation that the different ancestry-associated groups differ in some of their recombination parameters is very interesting: I can't recall seeing similar data before, beyond some general V gene expression level differences (typically thought to be a consequence of differing HLA allele distributions). However, having never performed such an ancestry-informative PCA before I have what may be naive concerns about it. For instance, of the ~400 people in the cohort, it seems that all are assigned to one of the groups, while I would presume that out of those 400 people there are likely some multiracial individuals or those who just fall out of the typical expected SNP distributions (which Fig 9 would suggest is the case). Some more discussion of this issue may be instructive for those like myself who have never run such an analysis. For example, are those individuals who are further from the center of their respective clusters - or perhaps those whose clustered ancestry group is different to their self-reported one - enriched in the outliers in Figs 7 and 8? Similarly, I'm curious as to how many individuals in each group end up assigned to another, and to which. These considerations seem especially important given the different sizes of the groups (with the 'Caucasian' associated group having hundreds of individuals, and the other groups mostly having fewer than ten), and the fact that the target clusters are themselves informed by the original input reported groupings. While obviously beyond the scope of this manuscript, it's interesting in itself to note that such an observation hasn't been made before. Presumably most TCRseq studies are not large or diverse enough to have detected such differences (even had people been looking)?

Thank you for your detailed suggestion. We have included a supplementary table (Table 1 - source data 1) containing the mapping between the original self-identified ancestry groups to the ancestry-informative PCA clusters. Also, we explored whether the outliers within Figure 7 were individuals whose self-identified ancestry group was different from their ancestry-informative PCA cluster. We largely found that this was not the case. For example, our comparison of the mean total N-insertions by PCA cluster (Figure 7) resulted in 15 total outliers (productive and non-productive analyses combined) which represented 10 individuals. Of these 10 individuals, six had matching self-identified ancestry and PCA cluster (both ``Caucasian''), one had self-identified American Black ancestry and was part of the ``African''-associated PCA cluster, one had self-identified Chinese ancestry and was part of the ``Asian''-associated PCA cluster, one had self-identified ``Caucasian'' ancestry and was part of the ``Hispanic''-associated PCA cluster, and one had an unknown self-identified ancestry and was part of the ``Caucasian''-associated PCA cluster. While it is possible that individuals with multiple-ancestries could fall out of the typical expected SNP distributions, we believe this is of little concern for our analyses shown in Figure 7. Also, to clarify our Figure 8 analyses, each data point within the figure depicts the minor allele frequency of a N-insertion associated SNP within the DNTT region, computed using only individuals within each PCA cluster. As such, the Figure 8 outliers represent SNPs, not individuals with potential multiple-ancestry.

Reviewer #3 (Recommendations for the authors):1. Regarding the datasets, full information regarding the donors from which the data have been used need to be summarized in a Table and couple of figures to identify possible (or no if this is the case) bias in terms of sex, age, ethnicity, influenza exposure in the days before the sample collection, CMV serotype, etc… Indeed, all these factors can influence the repertoire composition, and maybe some correction/normalization should be applied.

We have added Table 1 and Table 3 which contain cohort demographic information (e.g. sex, age, ancestry, and CMV serostatus) for the discovery and validation cohorts, respectively. While all of these factors may influence the repertoire composition, we have focussed on correcting factors which are known to affect both repertoire composition and SNP frequency directly. As such, for all of our analyses, we have incorporated ancestry-informative principal components as covariates in each model to correct for potential population-substructure-related effects which could inflate associations between each SNP and each repertoire feature of interest. Diagnostic statistics (genomic control statistic, λ) show that this bias correction is sufficient for all of our analyses. If additional confounding variables (e.g. age, sex, CMV serostatus) were still present within our analyses, we would expect these diagnostic statistics to indicate additional P-value inflation even after correcting for population substructure-related effects. Because this is not the case, we do not see evidence for additional confounding factors (e.g. age, sex, CMV serostatus).

2. On the source data table containing 9957 associations, 43 TRBV out the 66 mentioned showed significant associations with 1 to 508 SNPs. For several TRBV genes, the association with SNPs was significantly different according to the productive vs nonproductive origin of the TRBV in the dataset. Same for the 10 TRBJ (out of the 14 tested) showing significant associations with SNPs.First, the "significant association" column in Table 1 should reflect those results by indicating the number of V genes and J genes found to be associated with at least one SNP above the significance threshold.

We have updated Table 1 to include the number of V genes and J genes found to be associated with at least one SNP above the significance threshold, thanks.

Second, in the data source table we can see some V gene usage (and J gene usage as well) from nonproductive and productive rearrangements are associated with various number of SNPs. What is the overlap of the SNPs associated with each V according to its nonproductive/productive origin? In addition, a general comment on the use of nonproductive rearrangement data. Such "part" of the TCR repertoire is believed to reflect TCR generation independently of TCR selection. However, when analysed from blood TCR repertoires (like in this study), it is still unclear how much the nonproductive "repertoire" is biased by the fact that it is directly dependent on the productive, and therefore centrally/peripherally selected. Therefore, the variations associated at the V, J usage (as well as the trimming) may be biased by the immune history of every individual. Author should control on this possible bias or disregard the differences between productive vs. nonproductive sequences. The only dataset that could help address this question would be thymic cells at the different stage of differentiation.

Thank you for raising this concern. Since each non-productive rearrangement is sequenced due to the presence in the same T cell of a successful rearrangement that survived selection, we agree that it is possible that within-cell correlation between rearrangement events could imprint selection effects onto the non-productive repertoire.

However, it is commonly assumed that independence between the productive and non-productive recombination events breaks this linkage on the allele level (Robins, et al., DOI: 10.1126/scitranslmed.3001442; Murugan, et al., DOI: 10.1073/pnas.1212755109; Zvyagin, et al., DOI: 10.1073/pnas.1319389111; Rubelt, et al., DOI: 10.1038/ncomms11112; Pogorelyy, et al., DOI: 10.1073/pnas.1809642115).

Further, we are not aware of any evidence for a mechanism in which the two recombination events at the TCRbeta locus are strongly correlated. As evidence of independence, we find that many of the differences that we see between productive and non-productive sequences (e.g. absence of MHC associations for non-productive TCRs) are consistent with the interpretation of these sequences as "unselected". For example, all significant SNPs within the MHC region for V-gene usage in productive TCRs were not significant for non-productive TCRs, as expected. We found that many of the significant SNPs within the TRB region had similar association p-values between non-productive and productive TCRs. Notably, the majority of TRB region SNPs which were significant for productive TCRs and not significant for non-productive TCRs occurred for the usage of genes which have both productive and non-productive alleles; in these cases the SNP is likely a proxy for the productivity status of the allele on the same chromosome (see PMID 26596423). We have included a supplementary figure (Figure 1 —figure supplement 3) detailing the concordance between nonproductive and productive gene usage associations. This new analysis adds to the paper and we thank the reviewer for suggesting it.

Third, in the text it is indicated that "variation in the TCRB locus is most significantly associated with the expression of the gene TRBV28 for both the productive and nonproductive". Unless I missed something, this is also (maybe more) true for TRBV12-3, TRBV12-4 (known to be highly expressed in general) as well as TRBV24-1, accounting for respectively 353, 358 and 319 SNP associations compared with 424. Are those differences significant? Are you referring to TRBV28 for a particular reason (major overlap between SNPs associations between nonproductive and productive for instance or something else)? This should be clarified and detailed.

We have mentioned that "variation in the TCRB locus is most significantly associated with the expression of the gene TRBV28" in an effort to give some intuition for the strongest associations. We have clarified the wording of this sentence to indicate that expression of TRBV28 had the smallest, most significant association P-values for variation in the TCRB locus for both productive and non-productive TCRbeta chains (lines 143145).

We have also added the following sentence (lines 145-148) discussing which genes had the largest number of significant associations for variation in the TCRB locus:

"We identified the largest number of significant associations between variation in the TCRB locus and expression of the gene TRBV7-3 within productive TCRð›½ chains (232 significant associations) and the gene TRBJ12 within non-productive TCRð›½ chains (290 significant associations)."

3. On the association between HLADRB1 and TRBV10-3, the authors refers to two other studies that found such association however they omit to discuss the fact that on those studies they found in fact other TRBV gene usage to be much more associated with the HLADRB1, notably TRBV20-1 (Gao et al., 2019) which is not found from the Emerson dataset. Are the differences associated with the ethnicity (as Gao paper is mainly done on Asian population)? Maybe to provide some functional relationship of the associations, it could be of interest to analyze data from patients with AIDs, such as RA, SLE, Sjogren syndrome known to be associated with HLADRB1. Data from the Rossetti paper Rosseti et al., (Annals of Rheum Dis, 2017; TCRb data from Adaptive available online) as well as from the Liu et al. (Annals of Rheum Dis, 2019). For instance it could be of interest to determine whether in RA or SLE patients, a differential usage of TRBV10-3, TRBV20 compared to controls has been shown. Eventually, if DNA is still available, ensure the HLADRB1 genotype to correlate with the observations.

We thank the reviewer for these insightful observations. We have added a few sentences (lines 371-378) to the Discussion section to highlight that the specific associations identified between MHC locus variation and V-gene usage differed between the two previous studies (Sharon et al., 2016 and Gao et al., 2019) and our work reported here. We agree that the associations between HLA alleles and V-genes, and their variation across populations and in different disease contexts, constitute a fascinating topic for further investigation. However, we feel that deeper examination of these questions is outside the scope of the present manuscript: the high-resolution class I and class II HLA typing data for the Emerson cohort has been available since our 2018 *eLife* manuscript (DeWitt, et al; DOI: 10.7554/*eLife*.38358). In our view, the primary contribution of this manuscript is to connect the genome-wide genotyping data with TCR repertoire features. For example, this is the first description (to our knowledge) of SNPs in the TdT and Artemis loci that influence V(D)J recombination.

4. Regarding the SNPs association with the gene trimming and N-insertion numbers, interestingly the genes showing SNPs association with this TCR repertoire feature are definitely biologically linked. However, although the author distinguish the impact of the gene and on the trimming versus N-insertion, since the resulting repertoire analyzed is a post-selection repertoire, the observation are still bias by the selection effect. Moreover, it is also well known that shorted TCRs are more frequent in general, than long ones. In other word, authors should control for these bias and provide more evidence on the actual SNPs identified between the discovery and the validation cohorts.

Thank you for raising these concerns. As described above, while each nonproductive rearrangement may indirectly undergo selection along with its productive rearrangement counterpart, we are not aware of any evidence for a mechanism in which the productive and nonproductive recombination events are correlated. As such, we are assuming that independence between two recombination events breaks any sort of indirect selection linkage for the nonproductive rearrangement on the allele level. In regards to the validation analysis, the overlap between the discovery cohort and the validation cohort consisted of just two significant SNPs, one within the gene encoding the Artemis protein (DCLRE1C) and the other within the gene encoding the TdT protein (DNTT). We have described details regarding these SNPs and their associations within both the discovery and validation cohorts within Table 4.